# Precise control of alloying sites of bimetallic nanoclusters via surface motif exchange reaction

Qiaofeng Yao [1], Yan Feng[1,2], Victor Fung[3], Yong Yu [1], De-en Jiang[3], Jun Yang[2] & Jianping Xie [1]

Precise control of alloying sites has long been a challenging pursuit, yet little has been achieved for the atomic-level manipulation of metallic nanomaterials. Here we describe utilization of a surface motif exchange (SME) reaction to selectively replace the surface motifs of parent $[Ag_{44}(SR)_{30}]^{4-}$ (SR = thiolate) nanoparticles (NPs), leading to bimetallic NPs with well-defined molecular formula and atomically-controlled alloying sites in protecting shell. A systematic mass (and tandem mass) spectrometry analysis suggests that the SME reaction is an atomically precise displacement of SR–Ag(I)–SR-protecting modules of Ag NPs by the incoming SR–Au(I)–SR modules, giving rise to a core-shell $[Ag_{32}@Au_{12}(SR)_{30}]^{4-}$. Theoretical calculation suggests that the thermodynamically less favorable core-shell Ag@Au nanostructure is kinetically stabilized by the intermediate $Ag_{20}$ shell, preventing inward diffusion of the surface Au atoms. The delicate SME reaction opens a door to precisely control the alloying sites in the protecting shell of bimetallic NPs with broad utility.

[1] Department of Chemical and Biomolecular Engineering, National University of Singapore, 4 Engineering Drive 4, Singapore 117585, Singapore. [2] State Key Laboratory of Multiphase Complex Systems, Institute of Process Engineering, Chinese Academy of Sciences, 100190 Beijing, China. [3] Department of Chemistry, University of California, Riverside, CA 92521, USA. Qiaofeng Yao and Yan Feng contributed equally to this work. Correspondence and requests for materials should be addressed to D.-e.J. (email: de-en.jiang@ucr.edu) or to J.Y. (email: jyang@ipe.ac.cn) or to J.X. (email: chexiej@nus.edu.sg)

Mankind's use of alloy dates back to the Bronze Age, and thousands of years of development have shaped alloying into a fine and sophisticated craft of metallic materials for aesthetic, mechanic, and durability purposes. In the past decades, alloying has received intensive attention in metal nanomaterials research, as alloying presents unique opportunities for tuning physical and chemical properties (e.g., optical, catalytic, and magnetic) of metal nanoparticles (NPs)[1–7]. One emerging class of metal NPs is thiolate-protected metal nanoclusters (NCs) with particle size below 2 nm, often featuring well-defined molecular formula and structure[8–10]. Metal NCs in this sub-2-nm regime possess strong quantum confinement effects and exhibit some intriguing molecular-like properties, such as highest occupied molecular orbital (HOMO)–lowest unoccupied molecular orbital (LUMO) transition[11,12], quantized charging[13], intrinsic chirality[14,15], strong luminescence[16–20], and enhanced catalytic activity[21–23]. These properties are inherently (and sensitively) dictated by the size and composition of metal NCs[24–29]. Therefore, besides size control, composition engineering of metal NCs, such as alloying metal NCs with heteroatoms, provides an efficient way to diversify and tailor the physicochemical properties of metal NCs for practical applications in environment[30], energy[31–33], human health[34,35], and green catalysis[36,37].

In line with alloying metal NCs with heteroatoms, precise control of alloying sites in bimetallic NCs is a central task to realize their structure-/composition-dependent properties. In the past decade, intensive research efforts devoted to cluster chemistry have successfully produced a number of bimetallic and trimetallic NCs with atomic precision, leveraging on delicate co-reduction or galvanic replacement reactions[38–45]. For example, Yang et al.[38] have successfully synthesized $[Au_{12}@Ag_{20}@Ag_{12}(SR)_{30}]^{4-}$ NCs (SR denotes thiolate ligand) via co-reduction of a mixture of Au(I)–SR and Ag(I)–SR complexes in solution. In a separate study, Bootharaju et al.[46] developed an efficient way to dope a single-Au heteroatom into the center of $Ag_{13}$ core of $[Ag_{25}(SR)_{18}]^{-}$ by reacting AuClPPh$_3$ with $[Ag_{25}(SR)_{18}]^{-}$. In these successful attempts, Au heteroatoms were allocated in the core of alloy NCs, most probably due to ubiquitous involvement of the reduction reaction of Au(III)/Au(I) to Au(0). This replacement reaction is thermodynamically favorable since thiolate-protected metal NCs (i.e., $M_n(SR)_m$) typically adopt a core-shell structure of M(0)@M(I)–SR[11,47,48], where M(0) atoms would preferentially sit in the core of alloy NCs. As a result, Au@Ag NCs (or Au-core/Ag-shell NCs) were generally obtained via the co-reduction or galvanic replacement reactions. However, the same methods could not be used to obtain Ag@Au NCs (or Ag-core/Au-shell NCs), which are thermodynamically less favorable in solution. To address this challenging chemistry, a proposed replacement reaction should be able to largely inhibit the reduction reaction of Au(III)/Au(I) to Au(0).

Here we present a delicate and well-controlled surface motif exchange (SME) reaction, which could use the Au(I)–SR complexes in solution to precisely replace Ag(I)–SR-protecting motifs on the Ag NC surface, and allocate Au heteroatoms in the protecting shell of alloy NCs. The SME reaction revealed in this study is made possible by the insusceptibility of Au(I)–SR complexes towards reduction, and the structural similarity of Au(I)–SR complexes and Ag(I)–SR-protecting motifs of the parent Ag NCs. In particular, the strong Au(I)–S bond could largely stabilize Au (I) and prohibit the galvanic reduction of Au(I) by the Ag(0) core of Ag NCs, which could minimize the formation of Au@Ag NCs. More importantly, recent X-ray crystallography studies reveal that linear SR–(M(I)–SR)$_n$ structures are part of Ag(I)–SR-protecting motifs of thiolate-protected Ag NCs[43,48], which are also basic building modules of Au(I)–SR complexes. Such similarity in building module, together with close atomic radius of Au and

Ag, form the structural basis for the motif replacement reaction between Ag(I)–SR-protecting motifs and Au(I)–SR complexes. Our mass spectrometry (MS) and density functional theory (DFT) analyses also reveal that the SME is based on an atomically precise SR–M(I)–SR-protecting module, suggesting SME as an alternative reaction mechanism other than the reported metal[40,44] and ligand[49,50] exchange for composition engineering of metal NCs.

## Results

**Synthesis of $[Ag_{32}Au_{12}L_{30}]^{4-}$ NCs (L = SR or Cl).** Ag NCs are a promising class of functional materials with potential applications in the biomedical field, e.g., as efficient broad spectrum antimicrobial agents[51]. However, the relatively poor stability of Ag NCs in solution has limited their practical applications. Doping Ag NCs with Au heteroatoms is an efficient way to address this stability issue[38]; particularly, if the Au atoms could be placed on the surface of Ag NCs (i.e., formation of Ag@Au NCs), it could largely prohibit the interactions of Ag atoms inside the bimetallic NCs with possible oxidizing agents (e.g., oxygen) in the biological fluids. We chose $[Ag_{44}(SR)_{30}]^{4-}$ as a model Ag NC due to its well-understood cluster formula and crystal structure. The synthesis of $[Ag_{44}(SR)_{30}]^{4-}$ was according to a reported protocol[48,52], which involved reduction of Ag(I)–(p-MBA) complexes (p-MBA denotes para-mercaptobenzoic acid) by sodium borohydride (NaBH$_4$) in a highly alkaline semi-aqueous medium (see Methods for details). The as-prepared $[Ag_{44}(p-MBA)_{30}]^{4-}$ NCs could be readily dissolved in water or dimethylformamide (DMF) by either deprotonation or protonation in solution. The dissolved Ag NCs are dark-red in solution (Fig. 1a, right inset). Ultraviolet–visible (UV–vis) absorption spectrometry (Fig. 1a), polyacrylamide gel electrophoresis (PAGE, left inset of Fig. 1a), and electrospray ionization MS (ESI-MS, Fig. 1b) data are in good accordance to those of reported $[Ag_{44}(p-MBA)_{30}]^{4-}$ NCs[48,52], suggesting atomically precise $[Ag_{44}(p-MBA)_{30}]^{4-}$ has been obtained in high purity and large quantity sufficient for further motif exchange exploration. More details about the characterization of as-synthesized $[Ag_{44}(p-MBA)_{30}]^{4-}$ could be found in Supplementary Note 1.

The SME reaction was conducted by mixing protonated $[Ag_{44}(p-MBA)_{30}]^{4-}$ NCs with the pre-formed Au(I)–(p-MBA) complexes in DMF. In a typical reaction, 2 mL of DMF solution of protonated $[Ag_{44}(p-MBA)_{30}]^{4-}$ NCs (with a concentration of $[Ag_{44}] = 0.64$ mM) was mixed with a DMF dispersion of protonated Au(I)–(p-MBA) complexes (920 µL, [Au] = 16.7 mM and [p-MBA] = 50 mM), which were prepared by directly mixing HAuCl$_4$ with p-MBA in an aqueous solution followed by protonation in DMF. The reaction mixture was incubated in a thermomixer (650 rpm and 25 °C) for 2 h, and the color of reaction mixture changed from dark-red to pale-brown (Fig. 1c, right inset). The pale-brown solution was collected as raw product (with a typical yield of ~65% on Ag$_{44}$ NC basis, determined by inductively coupled plasma optical emission spectroscopy (ICP-OES)), which could be purified by cyclic centrifugation–dissolution for further characterization.

Accompanying the distinct color change of the NCs in solution are the changes of their UV–vis absorption spectra. As shown in Fig. 1c, four absorption features at 390 (peak), 490 (dominant peak), 620 (slight shoulder), and 735 nm (slight shoulder) were observed for the as-synthesized AgAu NCs. These absorption peaks are different from those of $[Ag_{44}(p-MBA)_{30}]^{4-}$ NCs (Fig. 1a), but they are somehow similar to those of $[Au_{12}@Ag_{32}(SR)_{30}]^{4-}$ NCs reported by Yang et al.[38]. To gain insights into the molecular formula of the as-synthesized AgAu NCs, we analyzed the samples by ESI-MS. As shown in Fig. 1d, only one group of cluster

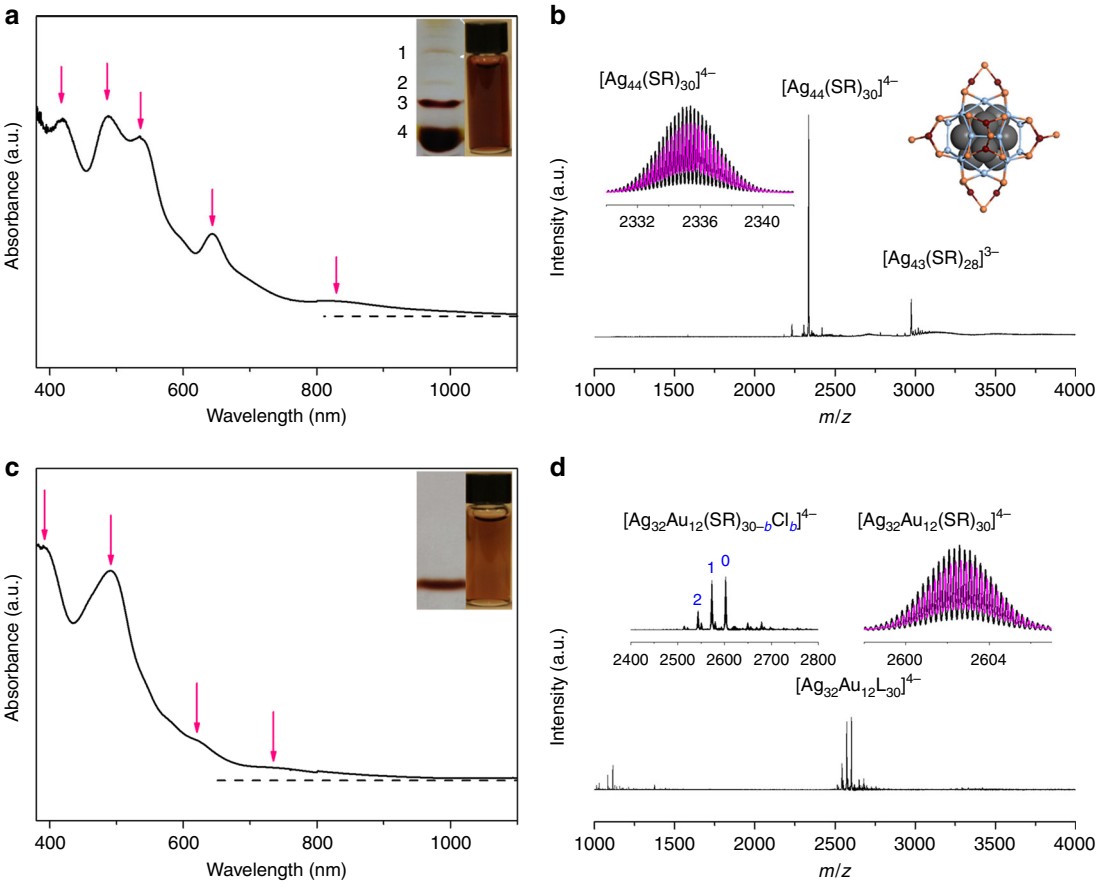

**Fig. 1** Synthesis of $Ag_{32}$@$Au_{12}$ from $Ag_{44}$ nanoclusters. **a, c** Ultraviolet–visible absorption and **b, d** electrospray ionization mass spectra of **a, b** $[Ag_{44}(SR)_{30}]^{4-}$ and **c, d** $[Ag_{32}Au_{12}(SR)_{30-b}Cl_b]^{4-}$ with $b = 0$–2 and SR denoting thiolate ligand. Insets in **a, c** are polyacrylamide gel electrophoresis results (left) and digital photos (right) of the corresponding nanoclusters. The zero absorbance is indicated by the dotted lines in **a, c**. The left inset in **b** shows the experimental (black line) and simulated (magenta line) isotope patterns of $[Ag_{44}(SR)_{30}]^{4-}$, and the right inset illustrates the crystal structure of $[Ag_{44}(SR)_{30}]^{4-}$ (drawn according to the reported structure[48]; color legend: Ag (gray/light blue/purple); S (orange); C, O, and H are omitted for clarity). The left inset in **d** is the zoom-in mass spectrum of $[Ag_{32}Au_{12}(SR)_{30-b}Cl_b]^{4-}$, and the right inset shows the experimental (black line) and simulated (magenta line) isotope patterns of $[Ag_{32}Au_{12}(SR)_{30}]^{4-}$

peaks centered at $m/z$ of 2602 could be found in a broad range of $m/z = 1000$–4000 (except minor peaks at $m/z < 1200$, which correspond to the singly charged small M(I)–SR complexes). This set of peaks are attributed to $[Ag_{32}Au_{12}L_{30}]^{4-}$ NCs (L = SR or Cl). The zoom-in spectrum at $m/z = 2400$–2800 (Fig. 1d, left inset) reveals that the cluster peaks are spaced regularly by $m/z = 29.51$, which could be transmitted to a mass difference of 118.04 ($= 29.51 \times 4$) Da by the 4– charge of cluster ion. Such mass difference complies well with the replacement of a $p$-MBA ligand (molecular weight, MW = 153.19) by a Cl ligand (MW = 35.5). Therefore, the as-synthesized AgAu NCs should be assigned as $[Ag_{32}Au_{12}(p\text{-MBA})_{30-b}Cl_b]^{4-}$ with $b = 0$–2, and the good accuracy of our assignment was evidenced by the superimposable experimental and calculated isotope patterns (Fig. 1d, right inset; and Supplementary Fig. 1). Of note, the high purity of the as-synthesized $[Ag_{32}Au_{12}L_{30}]^{4-}$ NCs was also evidenced in PAGE analysis (Fig. 1c, left inset), where only one band (in sharp contrast to the multiple bands of $[Ag_{44}(p\text{-MBA})_{30}]^{4-}$) was observed.

**Importance of eliminating unfavorable galvanic reactions.** The formation of high purity $[Ag_{32}Au_{12}L_{30}]^{4-}$ NCs should be attributed to a relatively mild (accomplished in ~1 h, see Supplementary Fig. 2 for time-evolution UV–vis absorption

spectra of the reaction solution) and controllable displacement reaction between the Ag(I)–SR-protecting motifs of $[Ag_{44}(SR)_{30}]^{4-}$ NCs and the incoming Au(I)–SR complexes in solution, making possible a complete elimination of the galvanic replacement reactions between Au(III) and the Ag(0) core in parent Ag NCs. The fast reaction kinetics (accomplished within 1 min) and the stoichiometry (1 Au(III) vs. 3 Ag(0)) of the galvanic replacement reaction between Au(III) salts and $[Ag_{44}(SR)_{30}]^{4-}$ NCs could easily destroy the $M_{44}S_{30}$ framework of the latter. This speculation was well supported by a series of control experiments. We first conducted alloying reaction by mixing $[Ag_{44}(SR)_{30}]^{4-}$ NCs with Au(III) salts in lieu of Au(I)–SR complexes, while keeping all other experimental conditions unvaried. In sharp contrast to $[Ag_{32}Au_{12}L_{30}]^{4-}$ NCs, AgAu NCs obtained by this way show a less distinct peak at ~490 nm in their UV–vis absorption spectrum (Supplementary Fig. 3), and six bands are seen in its PAGE gel (Supplementary Fig. 3, inset 3), suggesting polydisperse cluster size of the as-obtained AgAu NCs. A careful comparison of the ion mobility in gel (insets 1–3, Supplementary Fig. 3) and UV–vis absorption features (Supplementary Fig. 4) suggests that only Band 6 clusters possess a similar size and structure as $[Ag_{32}Au_{12}L_{30}]^{4-}$ and $[Ag_{44}(SR)_{30}]^{4-}$, while NCs in other bands most probably adopt different skeletons from $M_{44}L_{30}$. This set of UV–vis absorption and PAGE data are in good accordance to the ESI-MS analysis, which suggests AgAu

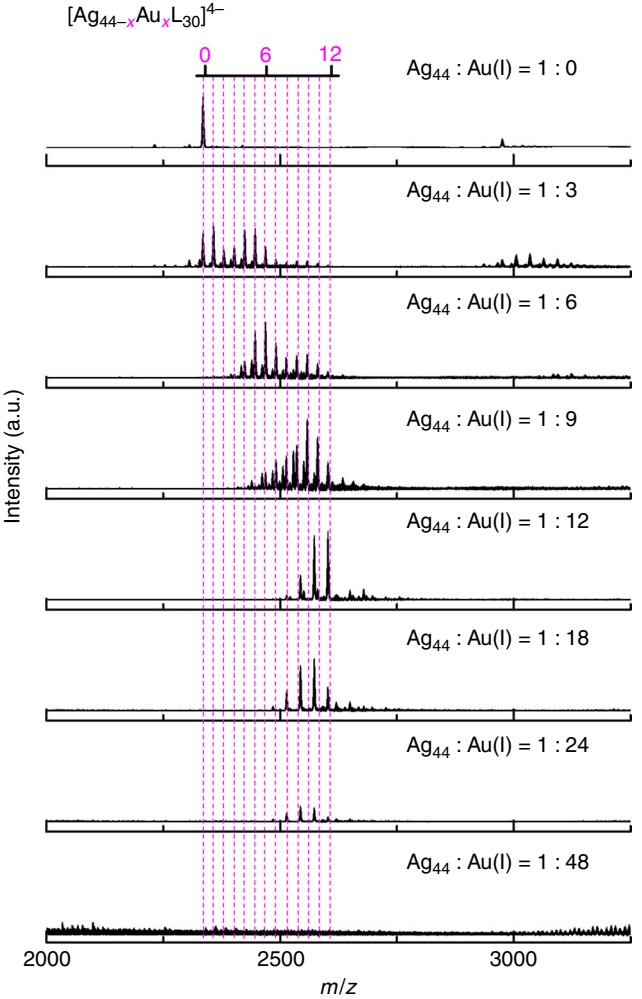

**Fig. 2** Reaction of $Ag_{44}$ nanoclusters with varied dose of Au(I)–SR complexes. Electrospray ionization mass spectra of $[Ag_{44-x}Au_xL_{30}]^{4-}$ nanoclusters synthesized at varied feeding ratios of $Ag_{44}$-to-Au(I), $R_{Ag44/Au(I)}$. The ligand (L) is thiolate (SR) or halogen (i.e., Cl). The dotted lines indicate the number of Au heteroatoms in each cluster

NCs produced by galvanic replacement reaction are a mixture of $[Au_xAg_{44-x}L_{30}]^{4-}$ ($x = 0$–9) and other-sized NCs (Supplementary Fig. 5).

In addition to the extreme case demonstrated above, where Au (III) salts were used as a sole Au source, we also carried out the alloying reactions in a mixture of Au(I)–SR complexes and Au (III) salts at varied molar Au(I)-to-Au(III) ratios ($R_{Au(I)/Au(III)}$). It has been widely accepted that Au(III) salts could react with thiols via the following balanced reaction (equation (1)):

$$n[AuCl_4]^- + 3n\,H - SR \rightarrow [Au(I) - SR]_n$$
$$+ n\,RS - SR + 3n\,H^+ + 4n\,Cl^- \quad (1)$$

According to this reaction, $R_{Au(I)/Au(III)}$ could be readily controlled by adjusting the molar feeding ratio of thiolate ligands (H-SR) and Au(III) salts ($R_{SR/Au}$) in an under-stoichiometry reaction. As shown in Supplementary Fig. 6, the AgAu NCs obtained at a higher $R_{SR/Au}$ possess more distinct absorption features of $[Ag_{12}Au_{32}L_{30}]^{4-}$ (Supplementary Fig. 6a) and a narrower size distribution (Supplementary Fig. 6b). This data suggests that the resultant AgAu NCs could be evolved into monodisperse $[Ag_{32}Au_{12}L_{30}]^{4-}$ by increasing $R_{SR/Au}$. It should be noted that monodisperse $[Ag_{32}Au_{12}L_{30}]^{4-}$ was formed at $R_{SR/Au} = 3$:1, which is in good accordance to the total conversion

stoichiometry of Au(III) into Au(I) by thiols (equation (1)). Further increasing $R_{SR/Au}$ to 4 would not change the value of $R_{Au(I)/Au(III)}$, and thus it has negligible impacts on the cluster size and size distribution of the resultant AgAu NCs. These data unambiguously validate the negative impacts of the galvanic reactions on the synthesis of monodisperse AgAu NCs. It also suggests that the proposed SME reaction has a better tolerance to excess thiolate ligands over excess Au(III) ions.

**Core-shell Ag@Au structure of $[Ag_{44-x}Au_xL_{30}]^{4-}$ NCs.** Shedding light on the alloying sites of AgAu NCs is another key objective of present study. To achieve this, we first examined the population of AgAu NCs (i.e., $x$ in $[Ag_{44-x}Au_xL_{30}]^{4-}$) formed at varied molar feeding ratio of $[Ag_{44}(SR)_{30}]^{4-}$ to Au(I) ($R_{Ag44/Au(I)}$), while keeping $R_{SR/Au}$ as a constant of 3:1. Figure 2 depicts ESI-MS spectra of AgAu NCs produced at various $R_{Ag44/Au(I)}$ ranging from ∞ (1:0) to 1:48, where the dotted lines serve as a visual guide of the varied $x$ values. The most notable feature at the early stage of alloying reaction is narrowing distribution of $x$ values and growing of max-population $x$ ($x_{max}$, denotes where the population of NCs reaches its maximum) with decreasing $R_{Ag44/Au(I)}$, where $(x, x_{max}) = (0–11, 1)$, $(4–12, 6)$, and $(5–12, 10)$ at $R_{Ag44/Au(I)} = 1$:3, 1:6, and 1:9, respectively. A close look at these MS spectra suggests that they consist of two sets of peaks, which could be well described by the formula of $[Ag_{44-x}Au_x(SR)_{30}]^{4-}$ and $[Ag_{44-x}Au_x(SR)_{29}Cl]^{4-}$ (Supplementary Fig. 7), respectively. It should be pointed out that no alloy NCs with $x > 12$ were observed, and the peaks with higher $m/z$ values than that of $[Ag_{32}Au_{12}(SR)_{30}]^{4-}$ recorded at $R_{Ag44/Au(I)} = 1$:9 should be assigned to Au(I)–SR complex-associated $[Ag_{44-x}Au_x(SR)_{30}]^{4-}$ ($x = 0$–10, Supplementary Fig. 8). Further decreasing $R_{Ag44/Au(I)}$ to 1:12 led to the formation of $[Ag_{32}Au_{12}(SR)_{30-b}Cl_b]^{4-}$ ($b = 0$–2 with a max-population $b$, $b_{max} = 0$; see detailed assignment in Fig. 1d). Bringing $R_{Ag44/Au(I)}$ down to 1:18 or 1:24 could not further boost the $x$ value in $[Ag_{44-x}Au_x(SR)_{30-b}Cl_b]^{4-}$, but it could elevate $b_{max}$ to 1 or 2, respectively (Supplementary Fig. 9). In association with the growing $b_{max}$ is the declining intensity of cluster ion peaks, which are completely diminished at an extreme $R_{Ag44/Au(I)}$ of 1:48 (bottom spectrum, Fig. 2).

Based on the above MS analyses, we can conclude that up to 12 Ag atoms in the frame of $Ag_{44}S_{30}$ could be substituted by Au heteroatoms; and below this extreme value, the $x$ values in $[Ag_{44-x}Au_xL_{30}]^{4-}$ NCs could be facilely modulated by changing the dose of Au(I)–SR complexes. The narrowing down distribution of $x$ values at the higher end of $R_{Ag44/Au(I)}$ (>1:12) should be attributed to the reduced freedom in alloying sites, as more alloying sites become occupied at decreased $R_{Ag44/Au(I)}$. On the other hand, the growing weight of Cl in the ligand shell (mirrored by increasing $b_{max}$) at the lower end of $R_{Ag44/Au(I)}$ (<1:12) is due to the ligand replacement reaction between SR ligand and free $Cl^-$, which is sourced from the ubiquitous Au(III) salt, $[AuCl_4]^-$. A number of recent achievements in X-ray crystallography of Au or AgAu NCs also corroborate the comparable capability of SR and Cl ligands of stabilizing the corresponding monometallic or bimetallic NCs, which further rationalizes the abovementioned ligand replacement of SR by Cl[53–55]. However, excess $Cl^-$ ions might further destroy the $M_{44}L_{30}$ framework; one good example is complete diminishing of cluster ion peaks and extensive formation of AgCl (Supplementary Fig. 10) when $R_{Ag44/Au(I)}$ is 1:48.

The up-limit of $x$ being 12 in $[Ag_{44-x}Au_xL_{30}]^{4-}$ NCs is an important clue towards mechanistic understandings on the proposed alloying reaction. Owing to marvelous contributions from two independent groups, Bigioni[48] and Zheng[38] groups, it has now become known that $[Ag_{44}(SR)_{30}]^{4-}$ adopts a three-layer structure (Fig. 1b, right inset), where a hollow icosahedral $Ag_{12}$

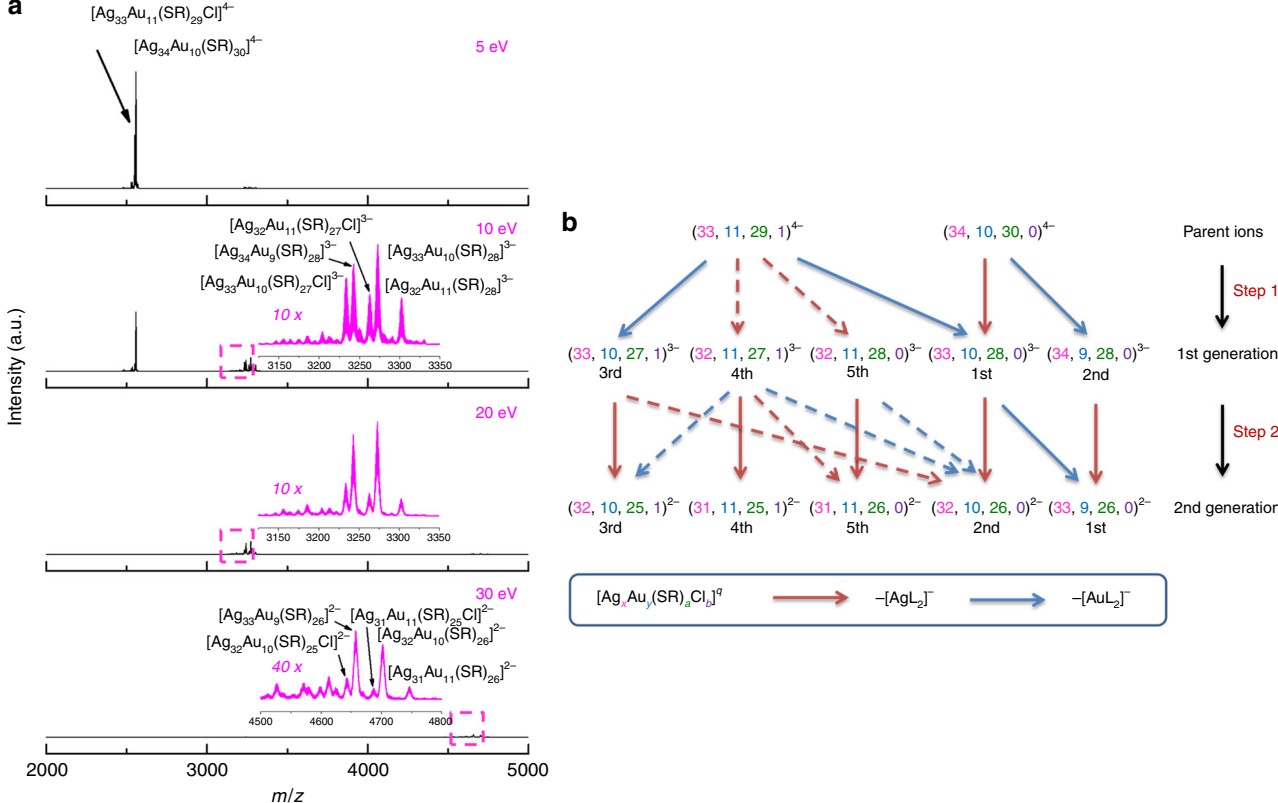

**Fig. 3** Fragmentation habit of Ag@Au nanoclusters. **a** Tandem mass spectra and **b** schematic illustration of fragmentation process of Ag-core/Au-shell $[Ag_{34}Au_{10}L_{30}]^{4-}$ and $[Ag_{33}Au_{11}L_{30}]^{4-}$ (marginal) ions (centered at $m/z = 2558$) obtained at different collision energies. The ligand (L) is thiolate (SR) or halogen (e.g., Cl), and $[Ag_xAu_y(SR)_aCl_b]^q$ nanoclusters are referred to as $(x, y, a, b)^q$ for simplicity. Insets of **a** are zoom-in spectra of the boxed area in the corresponding panels. The purple and blue arrows in **b** indicate fragmentation pathways by dissociation of $[AgL_2]^-$ and $[AuL_2]^-$, respectively; while the solid and dotted arrows show dominant and minor fragmentation pathways, respectively. The abundance sequence of each fragment cluster ion is indicated underneath in **b**

inner core (gray) is surrounded by a dodecahedral $Ag_{20}$ external core (light blue), and further capped by 6 mount-like $Ag_2(SR)_5$ motifs (12 Ag atoms colored in purple). The maximum 12 Au heteroatoms should be incorporated into either the $Ag_{12}$ inner core (core replacement) or those of $Ag_2(SR)_5$ motifs (motif replacement). In order to confirm which scenario (core or motif replacement) occurred in the alloying reaction, we turned to tandem MS (MS/MS) for more detailed insights.

MS/MS relies on monitoring the extensive fragmentation behaviors of targeting ions at varied collision energies, and it represents an important analytical technique widely used in identification and structure investigation of biomolecules and molecular-like clusters[56–58]. In a typical MS/MS analysis, the pristine ions with desired $m/z$ are selected in the first MS analysis (MS-1), and then subjected to a successive MS (MS-2) examination, where the fragmentation spectra of the pristine ions could be recorded under varied collision energies. As a demonstrative example, we chose $[Ag_{44}(SR)_{30}]^{4-}$ ion at $m/z = 2335$ in MS-1 as a model ion, and monitored their fragmentation spectra in a collision energy window of 5–30 eV. As shown in Supplementary Fig. 11, a higher collision energy produced smaller fragment ions (high $m/z$ end), which should be derived from the parent $[Ag_{44}(SR)_{30}]^{4-}$ ions by successive dissociation of single-negatively charged $[SR]^-$, $[Ag(SR)_2]^-$, and $[Ag_2(SR)_3]^-$ (low $m/z$ end). More details about the fragment cluster ions and their formation pathways can be found in Supplementary Fig. 12 and Supplementary Note 2. It should be noted that all these dissociated single-negatively charged species contain SR ligands, and only Ag atoms in the dodecahedral $Ag_{20}$ external core or $Ag_2(SR)_5$ motifs are coordinated with SR. We can then rationalize

that the fragmentations of thiolate-protected metal NCs in MS/MS might go through a stripping-off manner. Stable fragment cluster ions could then be formed by a successive dissociation of the SR-containing modules from $Ag_2(SR)_5$ surface motifs and the $Au_{20}$ external core they anchored on. Stripping-off these SR-containing modules would then cause exposure of the icosahedral $Ag_{12}$ inner core. Further fragmentations involving the icosahedral $Ag_{12}$ core would however ruin the Ag–S framework of $[Ag_{44}(SR)_{30}]^{4-}$, leading to an extinction of fragment cluster ions, which is exactly the case under a collision energy above 30 eV (Supplementary Fig. 13). It is worth noting that a similar fragmentation pathway induced by the stepwise dissociation of fractional surface protection motifs has also been observed in other metal NCs[57,58], suggesting the versatility of such fragmentation mechanism.

Based on the above understanding of the fragmentation preference of $[Ag_{44}(SR)_{30}]^{4-}$, we then carried out MS/MS analysis on our $[Ag_{44-x}Au_xL_{30}]^{4-}$ NCs, which could be referred to as $[Ag_xAu_y(SR)_aCl_b]^q$, or $(x, y, a, b)^q$ hereafter for a clear and easy presentation. As shown in Fig. 3a, bringing up the collision energy from 5 to 10 eV led to the formation of triple negatively charged first generation fragment cluster ions, i.e., $(33, 10, 27, 1)^{3-}$, $(32, 11, 27, 1)^{3-}$, $(32, 11, 28, 0)^{3-}$, $(33, 10, 28, 0)^{3-}$, and $(34, 9, 28, 0)^{3-}$; which were generated from the quadruple negatively charged parent ions $(34, 10, 30, 0)^{4-}$ and $(33, 11, 29, 1)^{4-}$ (marginal) centered at $m/z = 2558$, by dissociation of $[AuL_2]^-$ or $[AgL_2]^-$ modules (L = SR or Cl, Supplementary Fig. 14). The zoom-in MS/MS spectrum at $m/z = 3125$–3350 (Fig. 3a, inset in 10 eV spectrum) suggests a sequence of abundance of first generation fragment cluster ions as $(33, 10, 28, 0)^{3-} > (34, 9, 28, 0)^{3-} > (33, 10, 27, 1)^{3-} > (32, 11, 27, 1)^{3-} >$

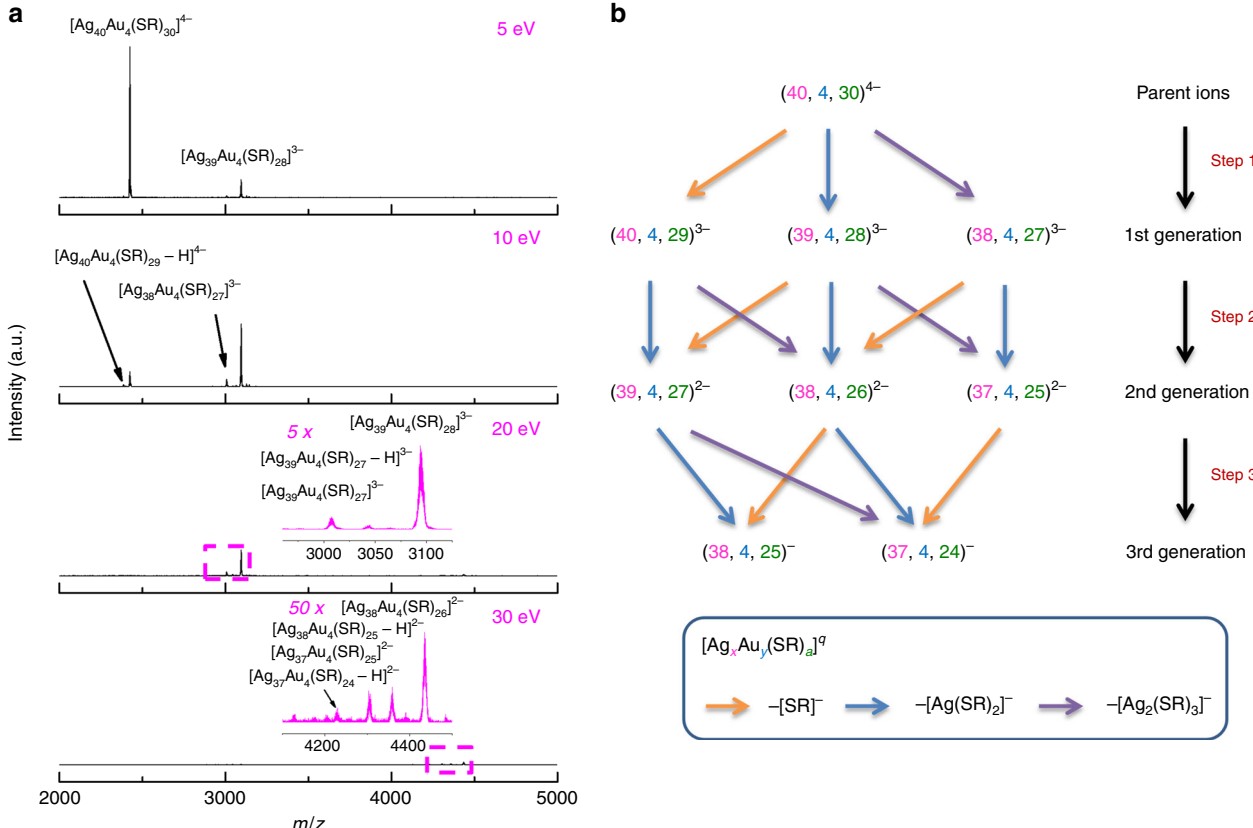

**Fig. 4** Fragmentation habit of Au@Ag nanoclusters. **a** Tandem mass spectra and **b** schematic illustration of fragmentation process of Au-core/Ag-shell $[Ag_{40}Au_4(SR)_{30}]^{4-}$ (centered at $m/z = 2424$; SR denotes thiolate ligand) obtained at different collision energies. Insets of **a** are zoom-in spectra of the boxed area in the corresponding panels. The orange, blue and purple arrows in **b** indicate the fragmentation pathways by dissociation of $SR^-$, $[Ag(SR)_2]^-$, and $[Ag_2(SR)_3]^-$, respectively

$(32, 11, 28, 0)^{3-}$ at the collision energy of 10 eV. This abundance sequence was made possible only by a fragmentation pattern illustrated in Step 1 of Fig. 3b, which features extensive dissociation of $[AuL_2]^-$ (blue arrows). Further increasing the collision energy to 20 eV could result in an exhaustion of parent cluster ions and the appearance of double negatively charged second generation fragment cluster ions. The peaks of these second generation fragment cluster ions would be further enhanced by diminishing of the first generation fragment cluster ions at higher collision energy (e.g., 30 eV). By a careful comparison of the formula of first and second generation fragment cluster ions (see Fig. 3b for detailed formula), we rationalized that those second generation fragment cluster ions could be derived from the corresponding first generation fragment ions by dissociating a $[AgL_2]^-$ (Fig. 3b, purple arrows in Step 2). The as-proposed fragmentation pathways are well supported by the almost unchanged abundance sequence of the corresponding ions in first and second generations (i.e., those paired by purple arrows) with a minor exception. The notable exception is the reversed abundance order in $(33, 10, 28, 0)^{3-} > (34, 9, 28, 0)^{3-}$ (first generation), and $(32, 10, 26, 0)^{2-} < (33, 9, 26, 0)^{2-}$ (second generation), which could be attributed to the partial decomposition of $(33, 10, 28, 0)^{3-}$ into $(33, 9, 26, 0)^{2-}$ by dissociation of $[AuL_2]^-$ (Fig. 3b, solid blue arrow in Step 2). Figure 3b suggests that extensive dissociation of $[AuL_2]^-$ (Step 1 in Fig. 3b) would preferentially occur prior to that of $[AgL_2]^-$ (Step 2 in Fig. 3b), which in the frame of stripping-off fragmentation scheme is a strong indication of incorporating Au atoms into the outmost surface modules. Reminding of the maximum number of the exchangeable Ag

atoms in $[Ag_{44}(SR)_{30}]^{4-}$ as 12, we can now conclude that the Au heteroatoms are located in the $M_2L_5$ motifs other than in the $M_{12}$ inner core of $[Ag_{44-x}Au_xL_{30}]^{4-}$ NCs.

To further confirm the Ag@Au structure of $[Ag_{44-x}Au_xL_{30}]^{4-}$ NCs produced by SME, we also compared their fragmentation habit with that of $[Ag_{44-x}Au_xL_{30}]^{4-}$ adopting conventional Au@Ag structure. The Au@Ag NCs were prepared by galvanic replacement reaction between $[Ag_{44}(SR)_{30}]^{4-}$ NCs and Au(III) salt, yielding a mixture of $[Ag_{44-x}Au_xL_{30}]^{4-}$ ($x = 0–9$, Supplementary Fig. 5). The most prominent $[Ag_{40}Au_4(SR)_{30}]^{4-}$ ion was then subjected to MS/MS analysis (Fig. 4a), and its fragmentation pathways are summarized in Fig. 4b. Intriguingly, Au-core/Ag-shell $[Ag_{44-x}Au_xL_{30}]^{4-}$ (Fig. 4b) exhibits the same fragmentation behavior as $[Ag_{44}L_{30}]^{4-}$ (Supplementary Fig. 12), where the fragment cluster ions are successively developed by dissociation of $L^-$, $[AgL_2]^-$, and $[Ag_2L_3]^-$ from the parent or last-generation fragment cluster ions. This is in sharp contrast to the preferential dissociation of $[AuL_2]^-$ in the fragmentation process of $[Ag_{44-x}Au_xL_{30}]^{4-}$ NCs generated by SME, unambiguously manifesting the Ag@Au structure of the latter.

**SME-based alloying reaction.** More direct evidences supporting the SME reaction mechanism are from the alloying reactions induced by Au(I)–SR′ complexes, where SR′ denotes a thiolate ligand other than those in the parent $[Ag_{44}(SR)_{30}]^{4-}$. Here we chose *para*-nitrothiophenol (*p*-NTP) as a foreign thiolate ligand, due to its structural similarity to *p*-MBA. As shown in Supplementary Fig. 15, the as-synthesized Ag@Au NCs by reacting $[Ag_{44}(p-MBA)_{30}]^{4-}$ with Au(I)–(*p*-NTP) complexes show a

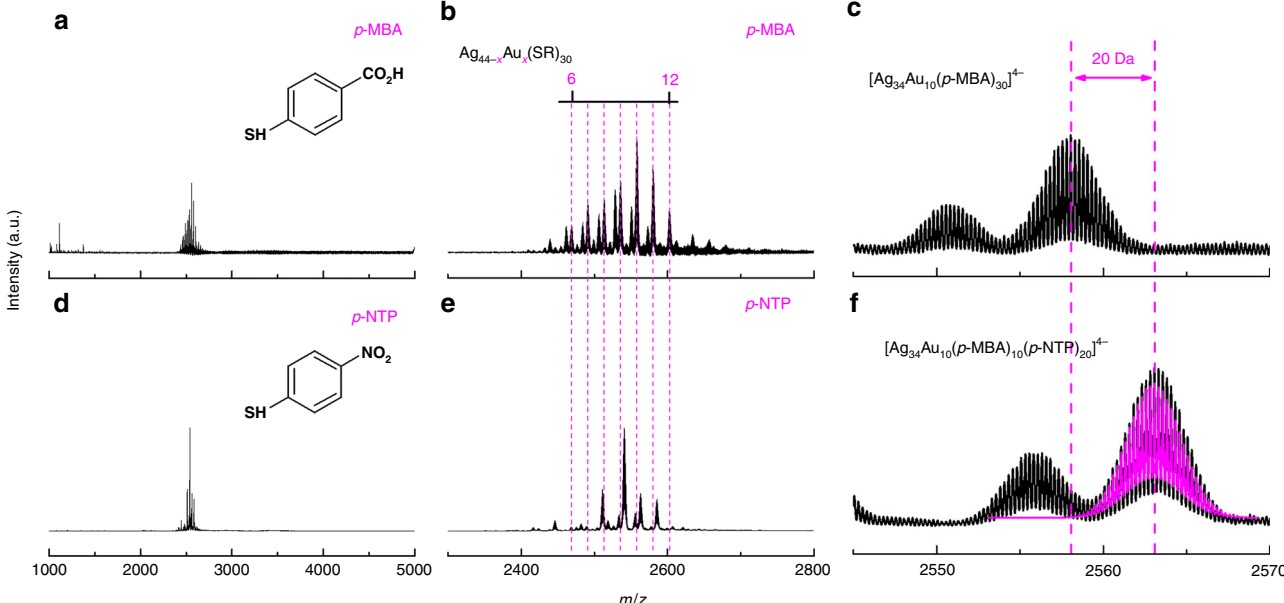

**Fig. 5** Ag@Au nanoclusters produced with different Au(I)–SR complexes. Electrospray ionization mass spectra of $[Ag_{44-x}Au_x(SR)_{30}]^{4-}$ nanoclusters (SR denotes thiolate ligand) synthesized by using **a–c** Au(I)-(p-MBA) complexes (p-MBA = para-mercaptobenzoic acid) and **d–f** Au(I)-(p-NTP) complexes (p-NTP = para-nitrothiophenol): **a, d** mass spectra in a broad $m/z$ range of 1000–5000; **b, e** zoom-in mass spectra in $m/z$ = 2200–2800; and **c, f** isotope patterns of $[Ag_{34}Au_{10}(SR)_{30}]^{4-}$. The dotted drop lines in **b, e** indicate the number of Au heteroatoms in each cluster. The magenta line in **f** is the simulated isotope pattern of $[Ag_{34}Au_{10}(p-MBA)_{10}(p-NTP)_{20}]^{4-}$

similar UV–vis absorption spectrum as that of Ag@Au NCs produced with Au(I)–(p-MBA) complexes, implying that these two NCs have a similar size and structure. ESI-MS data (Fig. 5a, d) indicates that $[Ag_{44-x}Au_x(p-MBA)_{30-b}(p-NTP)_b]^{4-}$ NCs ($x = 8–11$ with $x_{max} = 9$) were produced by using Au(I)–(p-NTP) complexes as Au precursor at $R_{Ag44/Au(I)}$ = 1:9, which is similar to $[Ag_{44-x}Au_x(p-MBA)_{30}]^{4-}$ ($x = 5–12$, with $x_{max} = 10$) obtained by replacing with Au(I)–(p-MBA) complexes at the same $R_{Ag44/Au(I)}$. A close comparison of the ESI-MS peaks (Fig. 5b, e) identified a minor discrepancy between the cluster peaks of $[Ag_{44-x}Au_x(p-MBA)_{30}]^{4-}$ and $[Ag_{44-x}Au_x(p-MBA)_{30-b}(p-NTP)_b]^{4-}$, which should be attributed to different MW of p-MBA (MW = 153.19) and p-NTP (MW = 154.17) ligands. For example, a mass difference of 20 Da (Fig. 5c, f) is found between $[Ag_{34}Au_{10}(p-MBA)_{30}]^{4-}$ and $[Ag_{34}Au_{10}(p-MBA)_{30-b}(p-NTP)_b]^{4-}$, which suggests 20 p-MBA ligands are displaced by p-NTP ligands when 10 Au heteroatoms are incorporated into $[Ag_{44}(SR)_{30}]^{4-}$ NCs. These delicate ESI-MS results imply that the alloying reaction occurs via replacing SR–Ag(I)–SR (a fraction of $Ag_2(SR)_5$ surface motifs) of the parent Ag NCs by the incoming Au(I)–SR complex.

To further figure out the key Au(I)–SR complex species involved in the SME reaction, we examined Au(I)–SR complex dispersion by ESI-MS. As shown in Supplementary Fig. 16, the dissolvable Au(I)–SR complex species are mostly $[Au_2(SR)_{3-b}Cl_b]^-$ ($b = 0–2$), and $[Au_2(SR)_2Cl]^-$ is the most prominent species. Therefore, the reaction that dominates the early stage of SME could be depicted by the following balanced reaction (equation (2)).

$$[Ag_{44}(SR)_{30}]^{4-} + [Au_2(SR')_2Cl^-] \rightarrow [Ag_{43}Au(SR)_{28}(SR')_2]^{4-}$$
$$+ [Au(SR)_2]^- + AgCl \qquad (2)$$

This reaction route is further supported by observation of $[Au(SR)_2]^-$ by-product at the low $m/z$ end of ESI-MS spectrum (Supplementary Fig. 17). With a sufficient supply of Au(I)–SR

complexes in solution, the SME reaction would then proceed to its completion through similar reaction route, and up to 12 (all) SR–Ag(I)–SR modules in $Ag_2(SR)_5$ motifs could be replaced by Au heteroatoms.

Based on the above reaction equation and the crystal structure of $[Ag_{44}(SR)_{30}]^{4-}$ (Fig. 6a), we have constructed an association–dissociation assisted SME mechanism for the alloying reaction, which is depicted in Fig. 6. Given $[Au_2(SR)_2Cl]^-$ as the incoming Au(I)–SR complex species, the high affinity of Cl to Ag would first initiate the adsorption of $[Au_2(SR)_2Cl]^-$ to a Ag atom in the $Ag_2(SR)_5$ protecting motif (Fig. 6b). Subsequent formation of Ag–Cl bond could cleave the S–Ag–S bond in the same $Ag_2(SR)_5$ motifs (Fig. 6c), leading to the formation of S-Au-S bond among two dangling S (from the parent cluster) and Au (from the complex). A subsequent dissociation of freshly formed AgCl and $[Au(SR)_2]^-$ would leave an open site on the $Ag_{20}$ external core, which could be capped by the remaining $[Au(SR)_2]^-$ residue from the $[Au_2(SR)_2Cl]^-$ (Fig. 6d). Of note, a similar pattern of SR bonding to the external core of metal NCs as the exchange site was also reported in previous ligand exchange studies[59–62]. The motif exchange reaction would then be completed by bonding the newly anchored SR–Au(I)–SR to the outmost dangling SR (Fig. 6e), followed by certain structure relaxations. As a net result, a SR–Ag(I)–SR module in the $Ag_2(SR)_5$ motif has been substituted by incoming SR–Au(I)–SR from the Au(I)–SR complexes in solution. The replacement of the remaining 11 SR–Ag(I)–SR could occur through a similar association–dissociation mechanism (Fig. 6f).

**Computational analysis on motif exchange reaction.** The known structure of $[Ag_{44}(SR)_{30}]^{4-}$ provides us a basis to gain further insights into the SME reaction from DFT modeling. The first question we tried to answer is the preference of the Ag sites for Au to replace. There are three layers of Ag atoms in $[Ag_{44}(SR)_{30}]^{4-}$, as shown in Fig. 7. We found that the substitution of the core Ag (i.e., from $Ag_{12}$ inner core) by Au is most preferred, followed by the surface Ag (i.e., from $Ag_2(SR)_5$ surface

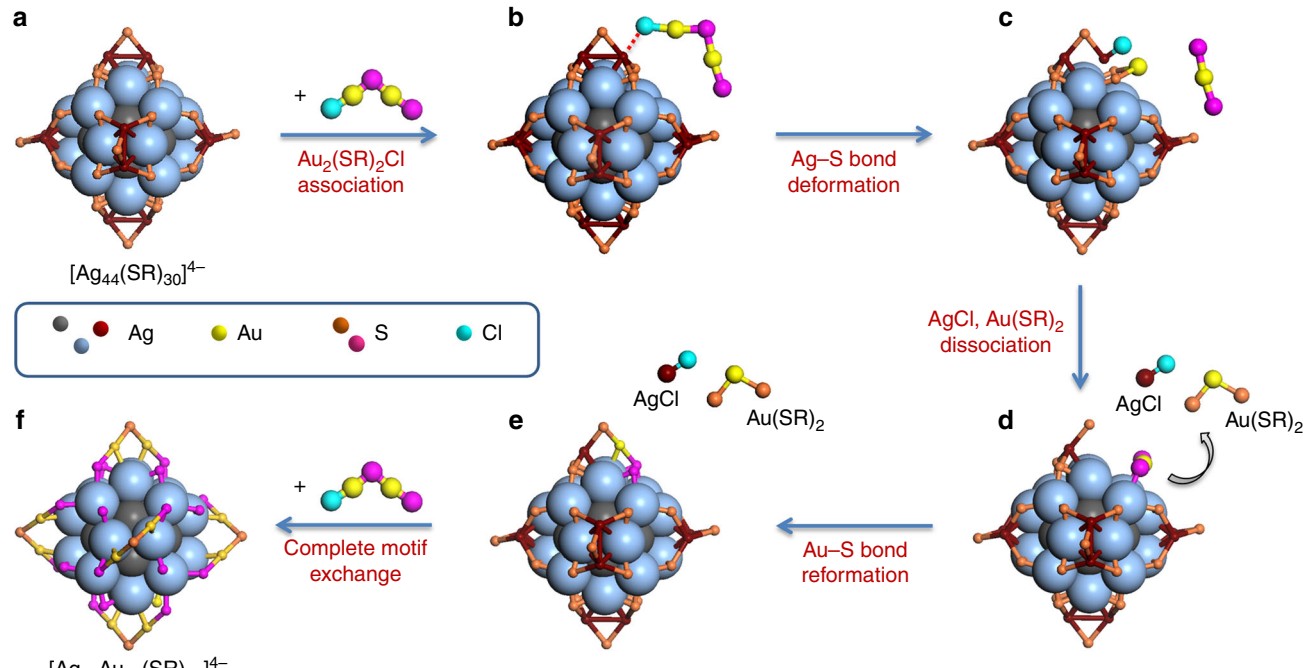

**Fig. 6** Schematic illustration of surface motif exchange reaction on $Ag_{44}$ nanoclusters. **a** Pristine $[Ag_{44}(SR)_{30}]^{4-}$; **b** association of $[Au_2(SR')_2Cl]^-$ to $[Ag_{44}(SR)_{30}]^{4-}$; **c** Ag-S bond deformation in $[Au_2(SR')_2Cl]^-$ associated $[Ag_{44}(SR)_{30}]^{4-}$; **d** dissociation of AgCl and $[Au(SR)_2]^-$; **e** $[Ag_{43}Au(SR)_{28}(SR')_2]^{4-}$ formed by exchange of one SR-Ag(I)-SR motif and **f** $[Ag_{32}Au_{12}(SR)_6(SR')_{24}]^{4-}$ produced by complete exchange of twelve SR-Ag(I)-SR motifs, where SR and SR' denote pristine and foreign thiolate ligands, respectively. For an easy and clear presentation, the Ag atoms in $Ag_{12}$ inner core and $Ag_{20}$ external core are shown as gray and light blue large balls, respectively; while the other atoms are shown as small dots (color legend: Ag (gray/light blue/purple), Au (yellow), S (orange/magenta), and Cl (light green)). The hydrocarbon tails and carboxylic groups of the protecting ligands are omitted

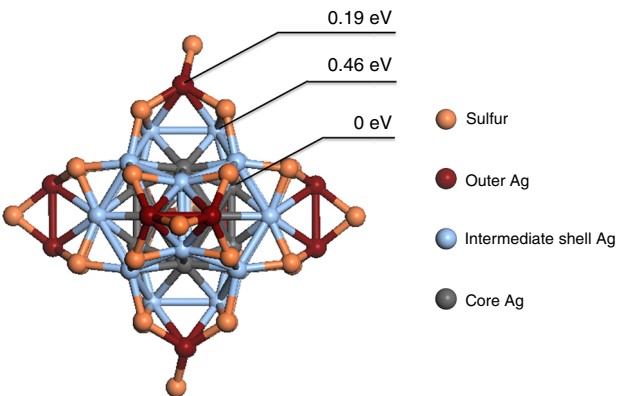

**Fig. 7** Energy diagram of Au replacement sites in $Ag_{44}$ nanoclusters. The Ag atoms of $[Ag_{44}(SR)_{30}]^{4-}$ (SR denotes thiolate ligand) are ordered into three general groups, the core Ag (gray), the intermediate shell Ag (light blue) and the outer shell Ag (purple). The sulfur (orange) atoms are shown, but the hydrocarbon tail and carboxylic ligand groups have been omitted for clarity. The relative energies for Au replacement on each of the three regions are labeled as shown

motifs) at 0.19 eV higher in energy, and then the intermediate shell Ag (i.e., from $Au_{20}$ external core) at 0.46 eV higher in energy. This relative energetics of substitution agrees with the experimental formation of Ag@Au NCs by the SME mechanism. A mild SME reaction on pre-formed $[Ag_{44}(SR)_{30}]^{4-}$ would first allow Au heteroatom to replace accessible Ag atoms (i.e., surface and intermediate shell Ag atoms), of which the surface Ag atoms are energetically favored. Upon the replacement of surface Ag atoms, the intermediate Ag shell would serve as a barrier layer for migration of surface Au atoms into the core, stabilizing the

thermodynamically less favored core-shell structure of Ag@Au NCs (against Au@Ag NCs).

To map out a pathway of the SME reaction from first principles is still too demanding and beyond the scope of present work. Therefore, we focused our computational efforts on the proposed Cl initiated S–Ag–S bond cleavage (Fig. 6b, c). Indeed, we found that Cl ligand can readily break the S–Ag–S bond and form a dangling Cl–Ag–S-complex attached to the cluster surface (Supplementary Fig. 18), without a barrier and with a favorable energy change of −0.06 eV. This computational evidence confirms the role of the Cl ligand in cleaving the surface S–Ag–S bonds that would facilitate the subsequent SME from Au(I)–SR complexes.

We next examined the likely structures of $[Ag_{44-x}Au_x(SR)_{30}]^{4-}$ NCs after some Ag atoms are substituted by Au atoms. Since the surface bonding of Au and Ag with thiolate ligands is different, it would be interesting to examine the changes to the structure on the surface. We first investigated the case of $x = 2$ and found that when the two Ag atoms in the outer shell $Ag_2(SR)_5$ motif are replaced by Au, a dimeric Au staple motif is formed across the diagonal of the $(SR)_4$ square (Fig. 8a); one can also see the Au–Ag bonding (Fig. 8b). Therefore, a reasonable model for $[Ag_{32}@Au_{12}L_{30}]^{4-}$ NC would be that the 12 Ag atoms in the six surface $Ag_2(SR)_5$ motifs are completely replaced by Au atoms. DFT-optimized structure of this model is shown in Fig. 8c and one can see that a highly symmetric structure is maintained. The DFT HOMO–LUMO gap of this cluster is 0.88 eV and close to that of $[Ag_{44}(SR)_{30}]^{4-}$ at 0.92 eV, indicating that the structure shown in Fig. 8c for the $[Ag_{32}@Au_{12}L_{30}]^{4-}$ NC is a viable model.

**Enhanced stability and scalability of core-shell Ag@Au NCs.** It is expected that the Au(I)–SR protection shell would enhance the stability of the as-synthesized core-shell Ag@Au NCs in solution. As evidenced by time-evolution UV–vis absorption spectra

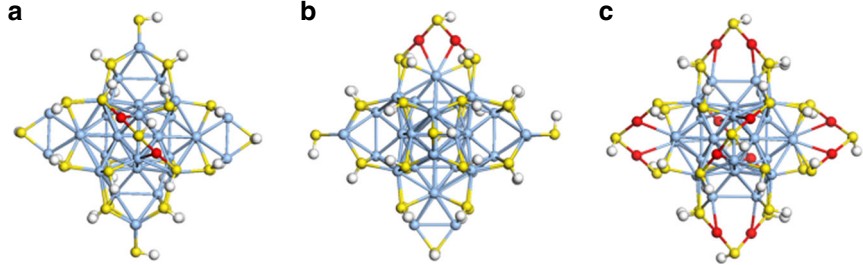

**Fig. 8** DFT-optimized structures of $Ag_{44-x}@Au_x$ nanoclusters. **a** Top and **b** side views of one outer shell $Ag_2(SR)_5$ motif of $[Ag_{44}(SR)_{30}]^{4-}$ (SR denotes thiolate ligand; color legend: Ag (light blue), S (yellow), and H (white)) substituted by Au (red) and the formation of the Au staple motif, and **c** proposed structure of $[Ag_{32}@Au_{12}(SR)_{30}]^{4-}$ by complete replacement of 12 outer shell Ag atoms. For ease of computation, the –SR is simplified as –SH

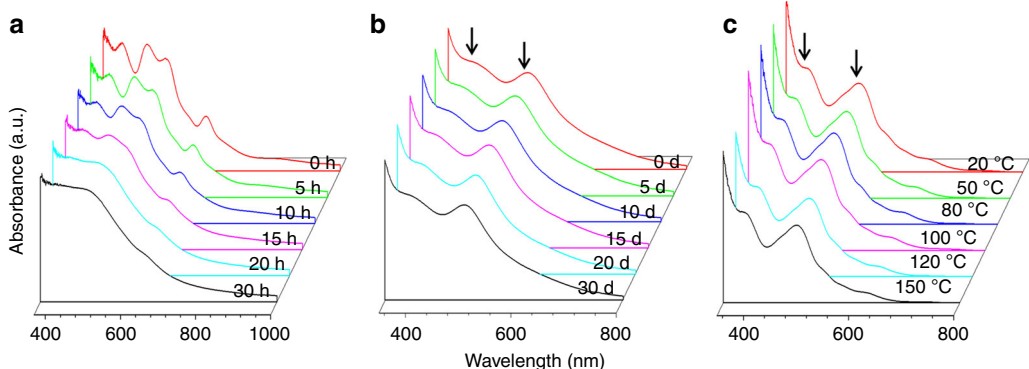

**Fig. 9** Enhanced stability of Ag@Au nanoclusters. Ultraviolet–visible absorption spectra of **a** $[Ag_{44}(SR)_{30}]^{4-}$ and **b**, **c** $[Ag_{32}Au_{12}L_{30}]^{4-}$ nanoclusters incubated in an elongated period (**a**, **b**, at room temperature) and at varied temperature (**c**, incubation time = 2 h). The ligand (L) is thiolate (SR) or halogen (e.g., Cl). The arrows in **b**, **c** indicate the absorption features of $[Ag_{32}Au_{12}L_{30}]^{4-}$ at 390 and 490 nm. Clusters in **a**, **b** were dissolved in 1 wt.% CsOH aqueous solution, while those in **c** were dissolved in dimethylformamide

(Fig. 9), the as-synthesized $[Ag_{32}Au_{12}L_{30}]^{4-}$ NCs were stable in aqueous solution at room temperature (25 °C) over an elongated time period (up to 30 days, Fig. 9b), while a similar incubation of $[Ag_{44}(SR)_{30}]^{4-}$ NCs in aqueous solution led to their extensive degradation within 2 days (Fig. 9a). In addition to long incubation time, $[Ag_{32}Au_{12}L_{30}]^{4-}$ NCs were also stable at high temperature. As shown in Fig. 9c, incubation of a DMF solution of $[Ag_{32}Au_{12}L_{30}]^{4-}$ NCs at elevated temperature (up to 150 °C) for 2 h only led to negligible changes in its UV–vis absorption spectrum. The robustness of Ag@Au NCs and the simplicity of their synthesis could facilitate the scale-up (Supplementary Fig. 19) of the SME-based synthetic chemistry. The delicate SME protocol is also facile and it can be extended to the systems of other coinage metal heteroatoms, such as copper (Cu; Supplementary Fig. 20).

## Discussion

In summary, we have developed an efficient SME reaction for production of core-shell Ag@Au NCs, with well-defined molecular formula and atomically precise alloying sites in the protecting shell. By a delicate interaction between the parent $[Ag_{44}(SR)_{30}]^{4-}$ NCs and the incoming Au(I)–SR complexes, $[Ag_{44-x}Au_xL_{30}]^{4-}$ ($x$ up to 12; L = SR or Cl) have been successfully synthesized. The alloying occurs through the exchange reaction between the atomically precise SR–Ag(I)–SR surface modules of $[Ag_{44}(SR)_{30}]^{4-}$ and the incoming SR–Au(I)–SR modules. This mild and well-controlled reaction is made possible by eliminating any possible galvanic reactions between the Ag(0) core of $[Ag_{44}(SR)_{30}]^{4-}$ and Au(III)/Au(I) species. Our DFT calculations further suggest that the formation of such thermodynamically less favorable core-shell Ag@Au structure is attributed to the diffusion barrier established by the $Ag_{20}$ external core of $[Ag_{44}(SR)_{30}]^{4-}$. This research is of interest not only because it provides a facile and scalable method to produce thermodynamically less favorable Ag@Au NCs, but also because it exemplifies that a delicate chemistry between the M(I)–SR complexes and thiolate-protected metal NCs could offer unique opportunities to precisely customize the structures of bimetallic NCs (e.g., composition and alloy sites) as well as their properties for both basic and applied explorations.

## Methods

**Materials**. Hydrogen tetrachloroaurate(III) trihydrate (HAuCl$_4$·3H$_2$O), copper(II) chloride (CuCl$_2$), *para*-mercaptobenzoic acid (*p*-MBA), *para*-nitrothiophenol (*p*-NTP), sodium borohydride (NaBH$_4$), cesium hydroxide (CsOH, 50 wt.% aqueous solution), and *N,N*-dimethylformamide (DMF) from Sigma Aldrich; silver nitrate (AgNO$_3$) and acetic acid (HOAc) from Merck; and ethanol from Fisher were used as-received without further purification. Ultrapure Millipore water (18.2 MΩ cm) was used in preparation of all aqueous solutions. All glassware were washed with aqua regia and rinsed with ethanol and ultrapure water before use.

**Synthesis of $[Ag_{44}(\textit{p}\text{-MBA})_{30}]^{4-}$ NCs**. $[Ag_{44}(\textit{p}\text{-MBA})_{30}]^{4-}$ NCs were prepared according to a reported protocol with some minor modifications[48,52]. In a typical synthesis, 3 mL of ethanolic solution of *p*-MBA (83 mM) was mixed with 5.25 mL of aqueous solution of AgNO$_3$ (23.8 mM) under vigorous stirring (1000 rpm), yielding a light-yellow suspension of Ag(I)–(*p*-MBA) complexes. The pH of reaction mixture was then brought up by adding in 200 μL of aqueous solution of CsOH (50 wt.%), which could immediately turn the cloudy suspension into clear solution. An aliquot of 2.25 mL of freshly prepared NaBH$_4$ solution (278 mM, in 0.1 M CsOH) was then dropped into the clear reaction solution to initiate the growth of $[Ag_{44}(\textit{p}\text{-MBA})_{30}]^{4-}$ NCs. The growth of $[Ag_{44}(\textit{p}\text{-MBA})_{30}]^{4-}$ could proceed to completion by stirring (1000 rpm) the reaction mixture at 50 °C for 12 h. A dark-red solution was collected as raw product at the end of this procedure.

The raw $[Ag_{44}(p\text{-}MBA)_{30}]^{4-}$ NCs were purified by cyclic centrifugation-redissolution. The as-prepared raw $[Ag_{44}(p\text{-}MBA)_{30}]^{4-}$ NC solution was first centrifuged at 10,000 rpm for 5 min to remove any likelihood insoluble impurity. The supernatant was then mixed with 10 mL of ethanol, followed by centrifugation at 10,000 rpm for 5 min. The precipitate was recovered and 5 mL of HOAc in DMF (30 vol.%) was added to preliminarily protonate the as-prepared $[Ag_{44}(p\text{-}MBA)_{30}]^{4-}$. The protonated $[Ag_{44}(p\text{-}MBA)_{30}]^{4-}$ NCs were recovered (as precipitate) by centrifugation at 10,000 rpm for 5 min. The recovered $[Ag_{44}(p\text{-}MBA)_{30}]^{4-}$ NCs were then redissolved in 5 mL of HOAc in DMF (10 vol.%), followed by a centrifugation at 10,000 rpm for 5 min in the presence of 10 mL of non-solvent toluene. After that, the precipitate (purified protonated $[Ag_{44}(p\text{-}MBA)_{30}]^{4-}$ NCs) was redissolved in DMF for further use. It is worth noting that the as-purified $[Ag_{44}(p\text{-}MBA)_{30}]^{4-}$ NCs could also be prepared in a deprotonated form via a similar cyclic centrifugation-redissolution method, simply by replacing DMF solution of HOAc by an aqueous solution of CsOH (1 wt.%) and using ethanol as universal non-solvent.

**Preparation of M(I)–SR Complexes**. Au(I)–SR complexes with varied feeding SR-to-Au(III) ratios ($R_{SR/Au}$) were prepared by mixing 2.4 mL of aqueous solution of thiolate ligands (e.g., $p$-MBA or $p$-NTP; at a calculated concentration) with an aqueous solution of HAuCl$_4$ (2 mL, 20 mM). For example, Au(I)–($p$-MBA) complexes with a $R_{SR/Au} = 3{:}1$ were synthesized by mixing aqueous solutions of $p$-MBA (2.4 mL, 50 mM) and HAuCl$_4$ (2 mL, 20 mM), followed by the addition of 44 μL of HOAc. The as-protonated Au(I)–($p$-MBA) complexes were obtained by centrifuging at 10,000 rpm for 5 min, and the precipitate was recovered and redispersed in 2.4 mL of DMF by sonication for 30 min. Cu(I)–SR complexes were prepared similarly by substituting HAuCl$_4$ with CuCl$_2$.

**Synthesis of AgAu NCs**. Ag@Au NCs were prepared by mixing DMF solutions of protonated $[Ag_{44}(SR)_{30}]^{4-}$ NCs and Au(I)–SR complexes. In a typical synthesis of $[Ag_{32}Au_{12}L_{30}]^{4-}$ NCs (L = SR or Cl), 920 μL of the as-prepared Au(I)–($p$-MBA) complexes ($R_{SR/Au} = 3{:}1$, dispersed in DMF) was added into 2 mL of protonated $[Ag_{44}(p\text{-}MBA)_{30}]^{4-}$ NCs (concentration of NCs = 0.64 mM). The reaction mixture was then incubated in a thermomixer (650 rpm, 25 °C) for 2 h. $[Ag_{32}Au_{12}L_{30}]^{4-}$ NCs were obtained as a pale-brown solution. $[Ag_{44-x}Au_xL_{30}]^{4-}$ NCs with varied $x$ values were synthesized using a similar procedure by adjusting the feeding amount of Au(I)–($p$-MBA) complexes ($R_{SR/Au} = 3{:}1$).

A similar procedure was used to prepare AgAu NCs at varied $R_{SR/Au}$ ratios. In particular, 920 μL of DMF dispersions of Au(I)–($p$-MBA) complexes prepared at varied $R_{SR/Au}$ ratios were mixed with 2 mL of protonated $[Ag_{44}(p\text{-}MBA)_{30}]^{4-}$, while keeping the other reaction conditions the same.

**Computational analysis**. The DFT calculations were performed using the Vienna ab initio Simulation Package[63,64]. The Perdew–Burke–Ernzerhof[65] functional of generalized-gradient approximation was used for the electron exchange and correlation. The electron-core interaction was described using the projector-augmented wave (PAW) method[66,67]. The kinetic energy cutoff was set to 450 eV for the plane wave basis set, and the Brillouin zone was sampled using the gamma point only. The Ag cluster placed in a 25 Å wide cubic cell to eliminate interactions in the periodic boundary conditions. The adsorption energies ($E_{ads}$) were calculated using $E_{ads} = E_{cluster+adsorbate} - (E_{cluster} + E_{adsorbate})$, where $E_{cluster+adsorbate}$, $E_{cluster}$, and $E_{adsorbate}$ are energies of cluster, adsorbate, and adsorbate-associated cluster, respectively. $E_{adsorbate}$ was computed by placing the adsorbate in a 15 Å wide cubic cell.

**Materials characterizations**. The alloying reaction was conducted in an Eppendorf Comfort thermomixer. UV–vis absorption spectra of the samples were obtained from a Shimadzu UV-1800 spectrometer. Cluster concentrations were measured by ICP-OES on a Thermo Scientific iCAP 6000. Scanning transmission electron microscopy and energy dispersive X-ray spectroscopy elemental mappings were recorded on a JEOL JEM 2010 microscope operating at 200 kV. ESI-MS analysis was carried out on a Bruker microTOF-Q system in negative ion mode. Detailed operating conditions of ESI-MS analysis are given as followings: source temperature/120 °C, dry gas flow rate/4 L per min, nebulizer pressure/0.4 bar, and capillary voltage/3.5 kV. In a typical ESI-MS analysis, 0.2 mL of DMF solution of NCs (with a NC concentration of ~0.64 mM) was injected with a flow rate of 3 μL per min. PAGE was carried out on a Bio-Rad Mini-Protean Tetra Cell system, with a stacking and resolving gel prepared from 4 and 30 wt.% acrylamide monomers, respectively. In a typical analytical gel (1.0 × 83 × 73 mm, 10 wells), 10 μL of aqueous NC solution containing 5 vol.% glycerol was loaded into each well, followed by running the gels at a fixed voltage of 200 V at 4 °C for 4 h.

**Data availability**. All relevant data are available from the corresponding author on request.

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

## Acknowledgements

We gratefully acknowledge financial supports from the Ministry of Education (Academic Research Grant R-279-000-481-112), Singapore, and the National Science Foundation of China (Grant No. 21573240) for our experimental investigation. DFT computation was supported by the U.S. Department of Energy, Office of Science, Basic Energy Sciences, Chemical Sciences, Geosciences, and Biosciences Division and used resources of the National Energy Research Scientific Computing Center, a DOE Office of Science User Facility supported by the Office of Science of the U.S. Department of Energy under Contract No. DE-AC02-05CH11231.

## Author contributions

Q.Y. and Y.F. contributed equally to this study. This project was supervised by J.X., D.-e.J. and J.Y. The experiments were designed by J.X., J.Y., Q.Y. and Y.F., and conducted by Q.Y., Y.F. and Y.Y. V.F. and D.-e.J. performed the DFT calculation. Discussion of results and manuscript preparation are group efforts of all authors.

## Additional information

**Competing interests:** The authors declare no competing financial interests.

