## [Peer Review File · Nature Communications]

Reviewers' comments:

Reviewer #1 (Remarks to the Author):

In this paper, the authors reports a surface motif exchange reaction to selectively replace the surface motifs of $[\text{Ag}_{44}(\text{SR})_{30}]_{4-}$ for the formation of $[\text{Ag}_{44-x}\text{Au}_x(\text{SR})_{30}]_{4-}$ nanoclusters. When Au(I)-SR was used for the exchange reaction, Au atoms in the obtained nanoclusters are located on their surface but not inside the core. Tandem MS (MS/MS) was used to evaluate the location of Au atoms. The results are quite unusual as compared to those reported in the literature, which should be interesting for publication in Nature Communications. However, before I can recommend its publication, the authors should address the following issues:

1. More direct evidences on the location of Au atoms should be provided. XAS (both XANES and EXAFS) should be powerful enough to offer these such evidences.
2. The authors claimed that the maximum number of the exchangeable Ag atoms in $[\text{Ag}_{44}(\text{SR})_{30}]_{4-}$ as 12. However, in Fig.2, there are peaks with the mass larger than $[\text{Ag}_{32}\text{Au}_{12}(\text{SR})_{30}]_{4-}$.
3. If Au atoms are located on the surface, it is surprising that $[\text{Ag}_{32}\text{Au}_{12}(\text{SR})_{30}]_{4-}$ exhibited much different from $[\text{Ag}_{44}(\text{SR})_{30}]_{4-}$. The authors should run some computations to answer the big difference. And the UV-vis profiles in Fig. 8b and 8c are different. Please explain too.
4. The structures of $[\text{Ag}_{32}\text{Au}_{12}(\text{SR})_{30}]_{4-}$ shown in Fig.5 and Fig.8 are different. Since surface Au should be linearly two coordinated, please draw Fig.5.
5. $[\text{Ag}_{44}(\text{SR})_{30}]_{4-}$ was claimed to be ultrastable. But Fig. 8a tells that the cluster was not very stable at room temperature. Please explain. Caused by light irradiation?
6. Can the developed surface motif exchange reaction extended to exchange other metal atoms onto the surface of $[\text{Ag}_{44}(\text{SR})_{30}]_{4-}$.

Reviewer #2 (Remarks to the Author):

In this manuscript the authors established a new strategy called Surface Motifs Exchange (SME) that they used to control the outer reaction sites of bimetallic nanoclusters with atomic precision. Engineering outer reaction sites is a hot topic in nanocluster chemistry and heterogeneous catalysis. Such strategies are key to enabling atomic-level control over catalysts, particularly nanoclusters and nanoparticles.

The engineering strategy presented by the authors requires an introduction of a heteroatom motif that has a similar structure to the original motif of the shell protecting the nanocluster. The authors demonstrate this concept on the $[\text{Ag}_{44}(\text{SR})_{30}]_{4-}$ nanocluster. The study also explains in detail the mechanism by which the monomeric staples of SR-Ag-SR on a $[\text{Ag}_{44}(\text{SR})_{30}]_{4-}$ nanocluster were successfully substituted by a RS-Au-RS complex to give a $[\text{Ag}_{32}\text{Au}_{12}(\text{SR})_{30}]_{4-}$ nanocluster.

Given the significance of the strategy developed in this work, I recommend the publication of this manuscript in Nature Communications after a minor revision that addresses the following comments:

- On page 18, the mechanism of displacement of the RS-Ag-AR motif the bonding of Au(I)_SR is not explained clearly.
- Characterizations that confirm the structure of the Au(I)-SR complex are missing.

Reviewer #3 (Remarks to the Author):

In this study the authors used Ag₄₄(SR)₃₀ NC as a model system to demonstrate a new mechanism based on surface motif exchange (SME) for nanoparticle surface alloying. The authors included details MS and UV-vis data and proposed the SME mechanism, which was also supported by results from DFT calculations. Indeed spatial control, in particular at the atomic level, is a great challenge and yet of fundamental and technological significance in nanoparticle structural engineering and functionalization. The work presented advances our understanding in such efforts, where deliberate control of the chemical reactivity (e.g., replacing active Au(III) with less active Au(I)-SR) of reaction precursors may be exploited for atomically precise replacement of nanoparticle surface motif. It is envisaged that this unique mechanism may be used as a generic, effective strategy for nanoparticle surface alloying. Overall, the work was carried out nicely and the authors did an excellent job in explaining the experimental results.

There are two issues that I hope the authors would address before the paper is accepted for publication in the journal. On page 20, the authors argued that the intermediate Ag shell served as a barrier layer for surface Au atoms reacting with inner core Ag. Should such a mechanism also exist when the NC reacts with Au(III)? In other words, using Au(III) as a precursor should also produce the same Ag@Au NC. The authors argued that Au(III) was far too active and the fast reaction kinetics might destroy the NC structure. Is it possible to carry out further DFT studies to address this issue?

Also, it will be great if some additional data such as XAS (EXAFS) can be included to directly unravel the Ag-Au structure.

Replies to reviewers' comments and descriptions of revisions made

Comments by Reviewer #1:

*In this paper, the authors reports a surface motif exchange reaction to selectively replace the surface motifs of $[Ag_{44}(SR)_{30}]^{4-}$ for the formation of $[Ag_{44-x}Au_x(SR)_{30}]^{4-}$ nanoclusters. When Au(I)-SR was used for the exchange reaction, Au atoms in the obtained nanoclusters are located on their surface but not inside the core. Tandem MS (MS/MS) was used to evaluate the location of Au atoms. The results are quite unusual as compared to those reported in the literature, which should be interesting for publication in *Nature Communications*. However, before I can recommend its publication, the authors should address the following issues:*

Reply: We are pleased and excited by the reviewer's positive acknowledgement on the novelty and significance of our study. Indeed, the mechanism and product (Ag-core/Au-shell or Ag@Au nanoclusters, NCs) of the surface motif exchange (SME) reaction presented in our study is distinctly different from those documented in alloying literature, and we believe it may open a new avenue for on-demand production of alloy nanomaterials at atomic precision. We believe the precise engineering of the alloying sites via the SME reaction, especially in the protecting shell of nanoparticles (NPs) will attract fundamental interests from heterogeneous readers of *Nature Communications*, stimulating more research activities in a diverse field of (alloying) metal chemistry, metal cluster chemistry, nanochemistry, and supramolecular chemistry. We would also like to thank the reviewer for his/her inspiring and constructive comments/suggestions, which have been taken into careful consideration in this revision. Please see below for a point-to-point response to the reviewer's specific comments/suggestions.

1. More direct evidences on the location of Au atoms should be provided. XAS (both XANES and EXAFS) should be powerful enough to offer these such evidences.

Reply: Thanks for this constructive suggestion. We totally agree with the reviewer that the manuscript could be improved by additional evidences that could unambiguously support the Ag@Au structure of alloy NCs produced in SME reaction, among which X-ray absorption spectroscopy (XAS) analysis and resolving the cluster structure via single-crystal X-ray diffraction (XRD) are two very powerful techniques. It has been widely accepted that XAS analysis is useful for revealing the structural features of metal NCs/NPs, organometallic complexes and biomolecules (*Science* **2016**, 352, 797; *Nat. Commun.* **2015**, 6, 7664; *Nat. Commun.* **2016**, 7, 10414; *J. Am. Chem. Soc.* **2015**, 137, 7027), and especially effective in probing the local coordination environment of central atoms. However, due to the limited time and access to synchrotron radiation sources, a systematic investigation of the location of Au heteroatoms in the Ag@Au NCs with XAS is beyond the scope of the present work. But it is definitely an important follow-up research topic to be pursued with significant efforts. On the other hand, we believe the additional tandem mass spectrometry (tandem MS or MS/MS) experiments that we have conducted in this revision (as described in the following paragraphs) could also address this issue.

In our original manuscript, we showed the tandem MS pattern of our present Ag@Au NCs. For structure comparison and contrast, here we present the tandem MS data for conventional Au@Ag NCs (Figures RL-1 and RL-2) prepared by simply mixing $[Ag_{44}(SR)_{30}]^{4-}$ NCs (SR denotes thiolate ligand) with Au(III) salt (i.e., HAuCl₄). The galvanic reduction of Au(III) salt by Ag(0) core of $[Ag_{44}(SR)_{30}]^{4-}$

could extensively yield Au(0) heteroatoms, preferentially incorporating into the core of resultant alloy NCs. As can be seen in Figure RL-1, the as-produced alloy NCs by galvanic replacement reaction are a mixture of $[\text{Ag}_{44-x}\text{Au}_x(\text{SR})_{30}]^{4-}$ ($x = 0-9$) and other-sized NCs. Such Au@Ag $[\text{Ag}_{44-x}\text{Au}_x(\text{SR})_{30}]^{4-}$ NCs were then analyzed by tandem mass spectrometry (MS/MS).

Figure RL-1 (Supplementary Figure 5 in the revised SI). Electrospray ionization mass spectrum of AgAu NCs formed by reacting $[\text{Ag}_{44}(\text{SR})_{30}]^{4-}$ with Au(III) salts (i.e., HAuCl_4). The inset shows zoom-in spectrum of $[\text{Ag}_{44-x}\text{Au}_x\text{L}_{30}]^{4-}$ ($\text{L} = \text{SR}$ or Cl) peaks. The $[\text{Ag}_{43-x}\text{Au}_x\text{L}_{28}]^{3-}$ is a common fragment ion of $[\text{Ag}_{44-x}\text{Au}_x\text{L}_{30}]^{4-}$, similar to the fragment ion of $[\text{Ag}_{43}\text{L}_{28}]^{4-}$ observed in the mass spectrum of $[\text{Ag}_{44}\text{L}_{30}]^{4-}$ (Figure 1b in main text). The asterisk peaks correspond to NCs with larger sizes, whose accurate formula could not be deduced due to a lack of isotope resolution.

The most prominent $[\text{Ag}_{40}\text{Au}_4(\text{SR})_{30}]^{4-}$ ion was then chosen as parent ion in the subsequent MS/MS analysis in a collision energy window of 5-30 eV. As shown in Figure RL-2, with elevating collision energy, 1st and 2nd generation of fragment cluster ions were generated by successive dissociation of single negatively charged SR^- , $[\text{Ag}(\text{SR})_2]^-$ and $[\text{Ag}_2(\text{SR})_3]^-$ from the parent or last-generation fragment cluster ions. Such fragmentation pathways (Figure RL-2b) are identical to those of $[\text{Ag}_{44}(\text{SR})_{30}]^{4-}$ NCs (Supplementary Figure 12), but they are distinctly different from those of Ag@Au $[\text{Ag}_{44-x}\text{Au}_x(\text{SR})_{30}]^{4-}$ produced by the SME reaction. In the case of Ag@Au NCs, a preferential dissociation of $[\text{Au}(\text{SR})_2]^-$ rather than $[\text{Ag}(\text{SR})_2]^-$ or other Ag-containing modules is featured in the fragmentation pathways (Figure 3). We believe that this comparison of MS/MS data unambiguously supports the Ag@Au structure model of $[\text{Ag}_{44-x}\text{Au}_x(\text{SR})_{30}]^{4-}$ generated by the SME reaction developed in the present study.

We have included these two figures and necessary discussions accordingly in the revised manuscript.

Figure RL-2 (Figure 4 in the revised manuscript). (a) Tandem mass spectra and (b) schematic illustration of fragmentation process of Au-core/Ag-shell $[Au_4Ag_{40}(SR)_{30}]^{4+}$ (centered at $m/z = 2450$) obtained at different collision energies. Insets in (a) are zoom-in spectra of the boxed area in the corresponding panels. The orange, blue and purple arrows in (b) indicate fragmentation pathways by dissociation of SR^- , $[Ag(SR)_2]^-$ and $[Ag_2(SR)_3]^-$, respectively.

Revisions:

Supplementary Information (SI), Page 6, Supplementary Figure 5:

Figure RL-1 is included in SI as Supplementary Figure 5.

Page 9, Lines 19-22:

“This set of UV-vis absorption and PAGE data is in good accordance to the ESI-MS analysis, which suggests AgAu NCs produced by galvanic replacement reaction are a mixture of $[Au_xAg_{44-x}L_{30}]^{4+}$ ($x = 0-9$) and other-sized NCs (Supplementary Fig. 5).”

Page 17, Lines 7-19:

“To further confirm the Ag@Au structure of $[Ag_{44-x}Au_xL_{30}]^{4+}$ NCs produced by SME, we also compared their fragmentation habit with that of $[Ag_{44-x}Au_xL_{30}]^{4+}$ adopting conventional Au@Ag structure. The Au@Ag NCs were prepared by galvanic replacement reaction between $[Ag_{44}(SR)_{30}]^{4+}$ NCs and Au(III) salt, yielding a mixture of $[Ag_{44-x}Au_xL_{30}]^{4+}$ ($x = 0-9$, see Supplementary Fig. 5). The most prominent $[Ag_{40}Au_4(SR)_{30}]^{4+}$ ion was then subjected to MS/MS analysis (Fig. 4a), and its fragmentation pathways are summarized in Fig. 4b. Intriguingly, Au-core/Ag-shell $[Ag_{44-x}Au_xL_{30}]^{4+}$ (Fig. 4b) exhibits the same fragmentation behavior as $[Ag_{44}L_{30}]^{4+}$ (Supplementary Fig. 12), where the fragment cluster ions are successively developed by dissociation of L^- , $[AgL_2]^-$, and $[Ag_2L_3]^-$ from the parent or last-generation fragment cluster ions. This is in sharp contrast to the preferential

dissociation of $[\text{AuL}_2]^-$ in the fragmentation process of $[\text{Ag}_{44-x}\text{Au}_x\text{L}_{30}]^{4-}$ NCs generated by SME, unambiguously manifesting the Ag@Au structure of the latter.”

Page 18, Figure 4:

Figure RL-2 is included in the main text as Figure 4.

2) The authors claimed that the maximum number of the exchangeable Ag atoms in $[\text{Ag}_{44}(\text{SR})_{30}]^{4-}$ as 12. However, in Fig. 2, there are peaks with the mass larger than $[\text{Ag}_{32}\text{Au}_{12}(\text{SR})_{30}]^{4-}$.

Reply: Thank you for this insightful comment. We are sorry for the confusion caused by less-detailed interpretation of MS data in the original submission. We have included the detailed analysis in the revised manuscript. It should be pointed out that there are no $[\text{Ag}_{44-x}\text{Au}_x\text{L}_{30}]^{4-}$ NCs with $x > 12$ captured in our MS analyses. According to the analysis of the isotope patterns, the peaks with m/z values higher than that of $[\text{Ag}_{32}\text{Au}_{12}(\text{SR})_{30}]^{4-}$ recorded in Figure 2 should be assigned to Au(I)-SR complex-associated $[\text{Ag}_{44-x}\text{Au}_x(\text{SR})_{30}]^{4-}$ ($x = 0-10$, Figure RL-3), which are most likely reaction intermediates in the SME reaction.

Figure RL-3 (Supplementary Figure 8 in the revised SI) (a) Zoom-in electro spray ionization mass spectrum of $[\text{Ag}_{44-x}\text{Au}_x\text{L}_{30}]^{4-}$ NCs synthesized at $R_{\text{Ag44}/\text{Au(I)}} = 1:9$, where peaks with mass higher than that of $[\text{Ag}_{32}\text{Au}_{12}(\text{SR})_{30}]^{4-}$ could be attributed to $[\text{Au}(\text{SR})\text{Cl}]^-$ associated $[\text{Ag}_{44-x}\text{Au}_x(\text{SR})_{30}]^{4-}$ ($x = 0-10$; $[\text{Ag}_{44-x}\text{Au}_x(\text{SR})_{30}\cdots\text{Au}(\text{SR})\text{Cl} + \text{H}]^{4+}$). (b) Experimental (black line) and simulated (magenta line) isotope patterns of $[\text{Ag}_{44}(\text{SR})_{30}\cdots\text{Au}(\text{SR})\text{Cl} + \text{H}]^{4+}$.

Revisions:

SI, Page 9, Supplementary Figure 8:

Figure RL-3 is included in the SI as Supplementary Figure 8.

Page 11, Lines 6-9:

“It should be pointed out that no alloy NCs with $x > 12$ were observed, and the peaks with higher m/z values than that of $[\text{Ag}_{32}\text{Au}_{12}(\text{SR})_{30}]^{4+}$ recorded at $R_{\text{Ag44}/\text{Au(I)}} = 1:9$ should be assigned to Au(I)-SR complex-associated $[\text{Ag}_{44-x}\text{Au}_x(\text{SR})_{30}]^{4+}$ ($x = 0-10$, Supplementary Fig. 8).”

3) *If Au atoms are located on the surface, it is surprising that $[\text{Ag}_{32}\text{Au}_{12}(\text{SR})_{30}]^{4+}$ exhibited much different from $[\text{Ag}_{44}(\text{SR})_{30}]^{4+}$. The authors should run some computations to answer the big difference. And the UV-vis profiles in Fig. 8b and 8c are different. Please explain too.*

Reply: Thank you for this insightful comment. We also noted the distinct perturbations in the ultraviolet-visible (UV-vis) absorption spectrum of NCs by Au heteroatom incorporation. The most prominent change is the almost diminished absorption at 645 nm after the SME reaction (Figures 1a and 1c). It has been documented (*Nat. Commun.* **2013**, 4, 2422) that the absorption at 645 nm exhibits combined characters of metal-to-metal (superatom 1D \rightarrow 1F) and ligand-to-metal (ligand \rightarrow 1F), which involve both metal atoms from the core and protecting shell. Therefore, it is expected that substitution of Ag atoms in either Ag_{32} core or outmost Ag_{12} shell (present study) could cause substantial changes to the absorption peak at 645 nm. Regarding computation of absorption features, the existing modeling method for optical absorption exhibits unsatisfactory inaccuracy in many cases, which may become considerable for distinguishing $[\text{Ag}_{32}\text{Au}_{12}(\text{SR})_{30}]^{4+}$ and $[\text{Ag}_{44}(\text{SR})_{30}]^{4+}$. For instance, the computed absorption spectrum of $[\text{Ag}_{44}(\text{SR})_{30}]^{4+}$ by time-dependent density functional theory (TDDFT) manifests an unimodal absorption peak at ~ 500 nm (*Nat. Commun.* **2013**, 4, 2422). Such absorption feature is distinctly different from the experimental bimodal peaks of $[\text{Ag}_{44}(\text{SR})_{30}]^{4+}$ (at 485 and 535 nm), but somehow close to that of the $[\text{Ag}_{32}\text{Au}_{12}(\text{SR})_{30}]^{4+}$ (unimodal peak at 490 nm) synthesized in the current study. Precisely reproducing and deciphering the absorption spectra of $[\text{Ag}_{44}(\text{SR})_{30}]^{4+}$ and $[\text{Ag}_{44-x}\text{Au}_x(\text{SR})_{30}]^{4+}$ by TDDFT will be a center of our follow-up computational efforts.

The absorption features in Figures 8b and 8c (Figures 9b and 9c in revised manuscript; only revised figure number will be quoted hereafter) are identical, where two prominent peaks were observed at 390 and 490 nm in both cases. We used slightly different x-axis scales in Figures 9b and 9c in our original submission, which is likely the dominant source of confusion. Another minor contributing factor is solvent effects. In particular, we dissolved $[\text{Ag}_{44}(\text{SR})_{30}]^{4+}$ (Figure 9a) and surface-doped $[\text{Ag}_{32}\text{Au}_{12}(\text{SR})_{30}]^{4+}$ (Figure 9b) in aqueous solution (1 wt.% CsOH was accommodated for good solubility and stability of the former), for a fair comparison of their stability in water, which is critical to many applications in aqueous environment (e.g., healthcare, catalysis, energy conversion, and environment monitoring). By contrast, DMF was chosen as the solvent in the thermostability test (Figure 9c) for a wider temperature window. We are sorry that we did not articulate well in the previous version, and we have re-drawn Figure 9b in an x-axis scale identical to that in Figure 9c. Visual guides of characteristic absorptions of $[\text{Ag}_{32}\text{Au}_{12}(\text{SR})_{30}]^{4+}$ at 390 and 490 nm and necessary experimental details have also been included in Figures 9b and 9c and their captions, respectively.

Revisions:

Page 25, Lines 1-5:

“As evidenced in time-evolution UV-vis absorption spectra (Fig. 9), the as-synthesized $[\text{Ag}_{32}\text{Au}_{12}\text{L}_{30}]^{4+}$ NCs were stable in aqueous solution at room temperature (25 °C) over an elongated time period (up to 30 days, Fig. 9b), while a similar incubation of $[\text{Ag}_{44}(\text{SR})_{30}]^{4+}$ NCs in aqueous solution led to their extensive degradation within 2 days (Fig. 9a).”

Page 25, Lines 6-8:

“As shown in Fig. 9c, incubation of a DMF solution of $[\text{Ag}_{32}\text{Au}_{12}\text{L}_{30}]^{4+}$ NCs at elevated temperature (up to 150 °C) for 2 h only led to negligible changes in its UV-vis absorption spectrum.”

Page 25, Figure 9:

Panel (b) has been re-drawn in an x-axis scale identical to that of panel (c). Visual guides of characteristic absorptions of $[\text{Ag}_{32}\text{Au}_{12}(\text{SR})_{30}]^{4+}$ have been added into panels (b) and (c).

Page 25, Caption of Figure 9:

“The arrows in (b, c) indicate the absorption features of $[\text{Ag}_{32}\text{Au}_{12}\text{L}_{30}]^{4+}$ at 390 and 490 nm. Clusters in (a, b) were dissolved in 1 wt.% CsOH aqueous solution, while those in (c) were dissolved in DMF.”

4) *The structures of $[\text{Ag}_{32}\text{Au}_{12}(\text{SR})_{30}]^{4+}$ shown in Fig.5 and Fig.8 are different. Since surface Au should be linearly two coordinated, please draw Fig.5.*

Reply: We have revised Figure 5 (Figure 6 in this revision) accordingly. We should also acknowledge this reviewer and the other two reviewers' constructive comments/suggestions, which have encouraged us to deepen our mechanistic understandings of the SME reaction. We have also included all the key information in the revised Figure 6. Thank you.

Revisions:

Pages 21, Figure 6:

The schematics have been re-drawn based on deepened mechanistic understandings of the SME reaction. A structural discrepancy regarding Au(I)-SR protecting motifs has also been corrected.

5) $[\text{Ag}_{44}(\text{SR})_{30}]^{4+}$ was claimed to be ultrastable. But Fig. 8a tells that the cluster was not very stable at room temperature. Please explain. Caused by light irradiation?

Reply: Thank you for this insightful comment. We agree with the reviewer that $[\text{Ag}_{44}(\text{SR})_{30}]^{4+}$ has been documented as one of the most stable atomically precise Ag NCs in the literature. However, it should be pointed out that the solution stability of $[\text{Ag}_{44}(\text{SR})_{30}]^{4+}$ is solvent dependent. For example, Bigioni and coworkers demonstrated that $[\text{Ag}_{44}(p\text{-MBA})_{30}]^{4+}$ ($p\text{-MBA} = para\text{-mercaptobenzoic acid}$) could be stabilized in dimethyl sulfoxide (DMSO) for at least 10 days (*Nature* **2013**, *501*, 399; *J. Phys. Chem. C* **2015**, *119*, 11238). However, the same group also observed immediate degradation of $[\text{Ag}_{44}(p\text{-$

MBA)₃₀]⁴⁻ in aqueous solution (*J. Phys. Chem. C* **2015**, *119*, 11238), which agrees well with the observations in our study. To evaluate their stability in aqueous solution, we dissolved [Ag₄₄(*p*-MBA)₃₀]⁴⁻ in 1 wt.% CsOH aqueous solution, where the alkaline condition could elongate the durability of [Ag₄₄(*p*-MBA)₃₀]⁴⁻ (from seconds to hours). The degradation mechanism is unclear at current stage, but we agree with the reviewer that the photochemical mechanism could be a plausible one. Other possible degradation pathways include oxidation by residual O₂ in water, perturbation of solvation layer, and inter-cluster collision (*J. Phys. Chem. C* **2015**, *119*, 11238). In addition to solvent effect, recent advances in cluster chemistry also suggest ligand effect as another equally important factor dictating the stability of [Ag₄₄(SR)₃₀]⁴⁻ NCs. For example, Bakr and coworkers demonstrated that the aqueous solution stability of [Ag₄₄(SR)₃₀]⁴⁻ could be significantly enhanced by using 5-mercapto-2-nitrobenzoic acid (MNBA) as protecting ligand, where negligible degradation was observed after 9-month incubation (*J. Mater. Chem. A* **2013**, *1*, 10148). Based on such good stability, the same group was able to develop a phase-transfer assisted ligand exchange strategies for facile surface engineering of [Ag₄₄(SR)₃₀]⁴⁻ NCs (*J. Am. Chem. Soc.* **2014**, *136*, 15865).

Revisions:

Page 25, Caption of Figure 9:

“The arrows in (b, c) indicate the absorption features of [Ag₃₂Au₁₂L₃₀]⁴⁻ at 390 and 490 nm. Clusters in (a, b) were dissolved in 1 wt.% CsOH aqueous solution, while those in (c) were dissolved in DMF.”

6) Can the developed surface motif exchange reaction extended to exchange other metal atoms onto the surface of [Ag₄₄(SR)₃₀]⁴⁻.

Reply: This is another constructive suggestion, and it prompted us to verify the versatility of the proposed SME reaction in other metal atoms. In particular, we have performed the SME reaction between [Ag₄₄(SR)₃₀]⁴⁻ and Cu(I)-SR complexes in solution. As shown in Figure RL-4, Cu heteroatoms could be incorporated into the frame of [Ag₄₄(SR)₃₀]⁴⁻ NCs by the SME reaction. In addition, the weight of Cu (i.e., *x* value) in the resultant [Ag_{44-x}Cu_x(SR)₃₀]⁴⁻ NCs could be tuned by changing the dose of Cu(I)-SR complexes. This data suggests that the proposed SME reaction is a generalized pathway to incorporate coinage metal heteroatoms into template [Ag₄₄(SR)₃₀]⁴⁻ NCs.

Revisions:

SI, Page 22, Supplementary Figure 20:

Figure RL-4 is included in SI as Supplementary Figure 20.

Page 25, Lines 8-12:

“The robustness of Ag@Au NCs and the simplicity of their synthesis could facilitate the scale-up (Supplementary Fig. 19) of the SME-based synthetic chemistry. The delicate SME protocol is also facile and it can be extended to the systems of other coinage metal heteroatoms, such as copper (Cu, Supplementary Fig. 20).”

Figure RL-4 (Supplementary Figure 20 in revised SI). Electrospray ionization mass spectra of $[\text{Ag}_{44-x}\text{Cu}_x(\text{SR})_{30}]^{4-}$ NCs synthesized by reacting $[\text{Ag}_{44}(\text{SR})_{30}]^{4-}$ with Cu(I)-SR complexes at varied feeding ratios of Ag₄₄-to-Cu(I), $R_{\text{Ag}_{44}/\text{Cu(I)}}$. The dotted lines indicate the number of Cu heteroatoms in each cluster.

Comments by Reviewer #2:

In this manuscript the authors established a new strategy called Surface Motifs Exchange (SME) that they used to control the outer reaction sites of bimetallic nanoclusters with atomic precision. Engineering outer reaction sites is a hot topic in nanocluster chemistry and heterogeneous catalysis. Such strategies are key to enabling atomic-level control over catalysts, particularly nanoclusters and nanoparticles.

The engineering strategy presented by the authors requires an introduction of a heteroatom motif that has a similar structure to the original motif of the shell protecting the nanocluster. The authors demonstrate this concept on the $[Ag_{44}(SR)_{30}]^{-4}$ nanocluster. The study also explains in detail the mechanism by which the monomeric staples of SR-Ag-SR on a $[Ag_{44}(SR)_{30}]^{-4}$ nanocluster were successfully substituted by a RS-Au-RS complex to give a $[Ag_{32}Au_{12}(SR)_{30}]^{-4}$ nanocluster.

Given the significance of the strategy developed in this work, I recommend the publication of this manuscript in Nature Communications after a minor revision that addresses the following comments:

Reply: We are glad that the reviewer finds the present SME strategy interesting and scientifically significant. Indeed, the SME reaction is built upon structural similarity between SR-Ag(I)-SR protecting module and the incoming Au(I)-SR complexes, together with a complete elimination of galvanic replacement reaction. By substituting atomically precise SR-Ag(I)-SR protecting modules of $[Ag_{44}(SR)_{30}]^{-4}$ by the incoming SR-Au(I)-SR modules, $[Ag_{44-x}Au_x(SR)_{30}]^{-4}$ NCs have been synthesized with up to 12 Au heteroatoms incorporating into the outmost protection shell. Such capability in controlling alloying sites in the protecting shell of NCs at atomic precision represents an important step towards atom-by-atom engineering metal nanomaterials for both fundamental and applied research. The detailed comments/suggestions of the reviewer have been fully considered in this revision, and a point-to-point response could be found in the following section.

1) *On page 18, the mechanism of displacement of the RS-Ag-AR motif the bonding of Au(I)-SR is not explained clearly.*

Reply: Thank you for this insightful suggestion. We are sorry that we didn't articulate well in the last submission. In this revision, we have provided more detailed and in-depth discussions on the kinetics and dynamics of the SME reaction. In particular, thanks to this reviewer's comment #2 (below), we have now identified the dominant Au(I)-SR complex species (Figure RL-5), i.e., $[Au_2(SR)_2Cl]^-$, which could initiate and fuel the SME reaction. We proposed the following balanced SME reaction:

where SR' denotes the foreign thiolate ligand. The reaction route detailed in the above equation was further supported by the capture of the by-product $[Au(SR)_2]^-$ in the SME reaction (Figure RL-6).

Figure RL-5 (Supplementary Figure 16). Electrospray ionization mass spectrum of Au(I)-SR complexes used in surface motif exchange reaction. The inset shows experimental (black line) and simulated (magenta line) isotope patterns of $[\text{Au}_2(\text{SR})_2\text{Cl}]^-$ peaks.

Figure RL-6 (Supplementary Figure 17). Electrospray ionization mass spectrum of Au(I)-SR complex by-product observed in surface motif replacement reaction. The inset shows experimental (black line) and simulated (magenta line) isotope patterns of $[\text{Au}(\text{SR})_2]^-$ peaks.

Based on the as-deduced atomically precise balanced equation together with the known crystal structure of $[\text{Ag}_{44}(\text{SR})_{30}]^{4-}$ (Figure RL-7a, *Nature* **2013**, 501, 399; *Nat. Commun.* **2013**, 4, 2422), we have constructed an association-dissociation assisted exchange mechanism for the SME reaction (Figure RL-7). Specifically, the high affinity of Cl to Ag would initiate the adsorption of $[\text{Au}_2(\text{SR})_2\text{Cl}]^-$ to a Ag atom in the $\text{Ag}_2(\text{SR})_5$ protecting motif (Figure RL-7b). Subsequent formation of Ag-Cl bond could cleave the S-Ag-S bond in the same $\text{Ag}_2(\text{SR})_5$ motifs (Figure RL-7c), leading to the formation of S-Au-S bond among two dangling S (from the parent cluster) and Au (from the complex). A subsequent dissociation of freshly formed AgCl and $[\text{Au}(\text{SR})_2]^-$ would leave an open site on the Ag_{20} external core, which could be capped by the remaining $[\text{Au}(\text{SR})_2]^-$ residue from the $[\text{Au}_2(\text{SR})_2\text{Cl}]^-$ (Figure RL-7d). The motif exchange reaction would then be completed by bonding the newly anchored SR-Au(I)-SR to the outmost dangling SR (Figure RL-7e), followed by certain structure relaxations. With a sufficient supply of Au(I)-SR complexes in solution, the SME reaction could proceed to completion, where all 12 Ag atoms in the outmost protecting shell of $[\text{Ag}_{44}(\text{SR})_{30}]^{4-}$ could be substituted by Au heteroatoms through a similar association-dissociation mechanism (Figure RL-7f).

Figure RL-7 (Figure 6 in the revised manuscript). Schematic illustration of surface motif exchange reaction on Ag_{44} nanoclusters. For an easy and clear presentation, the Ag atoms in Ag_{12} inner core and Ag_{20} external core are shown as grey and light blue large balls, respectively; while the other atoms are shown as small dots. The hydrocarbon tails and carboxylic groups of the protecting ligands are omitted.

Revisions:

SI, Page 18, Supplementary Figure 16:

The Figure RL-5 has been included in the SI as Supplementary Figure 16.

SI, Page 19, Supplementary Figure 17:

The Figure RL-6 has been included in the SI as Supplementary Figure 17.

Page 20, Line 1 – Page 21, Line 5:

“To further figure out the key Au(I)-SR complex species involved in the SME reaction, we examined Au(I)-SR complex dispersion by ESI-MS. As shown in Supplementary Fig. 16, the dissolvable Au(I)-SR complex species are mostly $[Au_2(SR)_{3-b}Cl_b]^-$ ($b = 0-2$), and $[Au_2(SR)_2Cl]^-$ is the most prominent species. Therefore, the reaction that dominates the early stage of SME could be depicted by the following balanced reaction (equation (2)).

This reaction route is further supported by observation of $[Au(SR)_2]^-$ by-product at the low m/z end of ESI-MS spectrum (Supplementary Fig. 17). With a sufficient supply of Au(I)-SR complexes in solution, the SME reaction would then proceed to its completion through this reaction route, and up to 12 (all) SR-Ag(I)-SR modules in $Ag_2(SR)_5$ motifs could be replaced by Au heteroatoms.”

Page 21, Figure 6:

The Figure RL-7 has been included in the main text as Figure 6.

Page 21, Line 6 – Page 22, Line 13:

“Based on the above reaction equation and the crystal structure of $[Ag_{44}(SR)_{30}]^{4-}$ (Fig. 6a), we have constructed an association-dissociation assisted SME mechanism for the alloying reaction, which

are depicted in Fig. 6. Given $[\text{Au}_2(\text{SR})_2\text{Cl}]^-$ as the incoming Au(I)-SR complex species, the high affinity of Cl to Ag would first initiate the adsorption of $[\text{Au}_2(\text{SR})_2\text{Cl}]^-$ to a Ag atom in the $\text{Ag}_2(\text{SR})_5$ protecting motif (Fig. 6b). Subsequent formation of Ag-Cl bond could cleave the S-Ag-S bond in the same $\text{Ag}_2(\text{SR})_5$ motifs (Fig. 6c), leading to the formation of S-Au-S bond among two dangling S (from the parent cluster) and Au (from the complex). A subsequent dissociation of freshly formed AgCl and $[\text{Au}(\text{SR})_2]^-$ would leave an open site on the Ag_{20} external core, which could be capped by the remaining $[\text{Au}(\text{SR})_2]^-$ residue from the $[\text{Au}_2(\text{SR})_2\text{Cl}]^-$ (Fig. 6d). Of note, a similar pattern of SR bonding to the external core of metal NCs as the exchange site was also reported in previous ligand exchange studies⁵⁹⁻⁶². The motif exchange reaction would then be completed by bonding the newly anchored SR-Au(I)-SR to the outmost dangling SR (Fig. 6e), followed by certain structure relaxations. As a net result, a SR-Ag(I)-SR module in the $\text{Ag}_2(\text{SR})_5$ motif has been substituted by incoming SR-Au(I)-SR from the Au(I)-SR complexes in solution. The replacement of the remaining 11 SR-Ag(I)-SR could occur through a similar association-dissociation mechanism (Fig. 6f).”

2) *Characterizations that confirm the structure of the Au(I)-SR complex are missing.*

Reply: Thank you for this constructive suggestion. As suggested, to further figure out the key Au(I)-SR complex species involved in the SME reaction, we examined Au(I)-SR complex dispersion by ESI-MS. As shown in Figure RL-5, the dissolvable Au(I)-SR complex species are mostly $[\text{Au}_2(\text{SR})_{3-b}\text{Cl}_b]^-$ ($b = 0-2$), and $[\text{Au}_2(\text{SR})_2\text{Cl}]^-$ is the most prominent species. The known identity of the dominant Au(I)-SR species has spurred substantial improvements in mechanistic understanding of the SME reaction (detailed in our response to comment #1 above).

Comments by Reviewer #3:

In this study the authors used $\text{Ag}_{44}(\text{SR})_{30}$ NC as a model system to demonstrate a new mechanism based on surface motif exchange (SME) for nanoparticle surface alloying. The authors included details MS and UV-vis data and proposed the SME mechanism, which was also supported by results from DFT calculations. Indeed spatial control, in particular at the atomic level, is a great challenge and yet of fundamental and technological significance in nanoparticle structural engineering and functionalization. The work presented advances our understanding in such efforts, where deliberate control of the chemical reactivity (e.g., replacing active Au(III) with less active Au(I)-SR) of reaction precursors may be exploited for atomically precise replacement of nanoparticle surface motif. It is envisaged that this unique mechanism may be used as a generic, effective strategy for nanoparticle surface alloying. Overall, the work was carried out nicely and the authors did an excellent job in explaining the experimental results.

Reply: We are glad that the reviewer finds our work interesting, significant and well-executed. As the reviewer spotted, surface engineering is a rising topic in the fundamental and applied research of nanostructured materials (*Science* **2016**, 354, 1580; *Nature Commun.* **2015**, 6, 7664; *Nat. Commun.* **2013**, 4, 1454; *J. Am. Chem. Soc.* **2013**, 135, 4946; *J. Am. Chem. Soc.* **2014**, 136, 15865). Our study could advance the synthetic and mechanistic understanding on this topic at atomic precision. By replacing commonly used Au(III) salt with less reactive Au(I)-SR complexes, we are able to eliminate the unfavorable galvanic replacement reaction between Au(III) and the Ag(0) core of $[\text{Ag}_{44}(\text{SR})_{30}]^{4+}$, which could induce a mild and quantitative SME reaction of $[\text{Ag}_{44}(\text{SR})_{30}]^{4+}$ with the incoming Au(I)-SR complexes. Such mild and controllable SME reaction could generate $[\text{Ag}_{44-x}\text{Au}_x(\text{SR})_{30}]^{4+}$ with well-defined formula and molecule structure, by precisely allocating Au heteroatoms in the protecting shell.

1) There are two issues that I hope the authors would address before the paper is accepted for publication in the journal. On page 20, the authors argued that the intermediate Ag shell served as a barrier layer for surface Au atoms reacting with inner core Ag. Should such a mechanism also exist when the NC reacts with Au(III)? In other words, using Au(III) as a precursor should also produce the same Ag@Au NC. The authors argued that Au(III) was far too active and the fast reaction kinetics might destroy the NC structure. Is it possible to carry out further DFT studies to address this issue?

Reply: We are thankful for the reviewer's constructive suggestions/comments. As suggested by the reviewer, we conducted structural analysis for the alloy NCs produced by galvanic replacement reaction with Au(III) salt. As shown in Figure RL-1 (or Supplementary Figure 5 in the revised SI, copied below for easy reading), such galvanic replacement reaction generates a mixture of alloy NCs with varied sizes, among which $[\text{Ag}_{44-x}\text{Au}_x(\text{SR})_{30}]^{4+}$ is an identifiable component. We then analyzed the most prominent $[\text{Ag}_{40}\text{Au}_4(\text{SR})_{30}]^{4+}$ ions by MS/MS. As depicted in Figure RL-2 (or Figure 4 in the revised manuscript, copied below), the fragmentation of $[\text{Ag}_{40}\text{Au}_4(\text{SR})_{30}]^{4+}$ occurs with elevating collision energy via successive dissociation of single negatively charged SR^- , $[\text{Ag}(\text{SR})_2]^-$, and $[\text{Ag}_2(\text{SR})_3]^-$. The absence of fragmentation pathway by dissociation of Au-containing surface module, which is characteristic of Ag@Au NCs, inherently implies their Au@Ag structure. Moreover, a careful comparison of Figure RL-2b with Supplementary Figure 12 suggests an identical fragmentation pathway of galvanically-generated $[\text{Ag}_{44-x}\text{Au}_x(\text{SR})_{30}]^{4+}$ and $[\text{Ag}_{44}(\text{SR})_{30}]^{4+}$, corroborating the Au@Ag structure of the former. As mapping out detailed reaction pathways for the galvanic replacement reaction between $[\text{Ag}_{44}(\text{SR})_{30}]^{4+}$ and Au(III) by DFT calculation is still too demanding and beyond the scope of this work, we may pursue it in a separate work in near future.

Figure RL-1 (Supplementary Figure 5 in the revised SI). Electrospray ionization mass spectrum of AgAu NCs formed by reacting $[\text{Ag}_{44}(\text{SR})_{30}]^{4-}$ with Au(III) salts (i.e., HAuCl_4). The inset shows zoom-in spectrum of $[\text{Ag}_{44-x}\text{Au}_x\text{L}_{30}]^{4-}$ ($\text{L} = \text{SR}$ or Cl) peaks. The $[\text{Ag}_{43-x}\text{Au}_x\text{L}_{28}]^{3-}$ is a common fragment ion of $[\text{Ag}_{44-x}\text{Au}_x\text{L}_{30}]^{4-}$, similar to the fragment ion of $[\text{Ag}_{43}\text{L}_{28}]^{4-}$ observed in the mass spectrum of $[\text{Ag}_{44}\text{L}_{30}]^{4-}$ (Figure 1b in main text). The asterisk peaks correspond to NCs with larger sizes, whose accurate formula could not be deduced due to a lack of isotope resolution.

Figure RL-2 (Figure 4 in the revised manuscript). (a) Tandem mass spectra and (b) schematic illustration of fragmentation process of Au-core/Ag-shell $[\text{Au}_4\text{Ag}_{40}(\text{SR})_{30}]^{4-}$ (centered at $m/z = 2450$) obtained at different collision energies. Insets in (a) are zoom-in spectra of boxed area in corresponding panels. The orange, blue and purple arrows in (b) indicate fragmentation pathways by dissociation of SR^- , $[\text{Ag}(\text{SR})_2]^-$ and $[\text{Ag}_2(\text{SR})_3]^-$, respectively.

Revisions:

SI, Page 6, Supplementary Figure 5:

Figure RL-1 is included in SI as Supplementary Figure 5.

Page 9, Lines 19-22:

“This set of UV-vis absorption and PAGE data is in good accordance to the ESI-MS analysis, which suggests AgAu NCs produced by galvanic replacement reaction are a mixture of $[\text{Au}_x\text{Ag}_{44-x}\text{L}_{30}]^{4-}$ ($x = 0-9$) and other-sized NCs (Supplementary Fig. 5).”

Page 17, Lines 7-19:

“To further confirm the Ag@Au structure of $[\text{Ag}_{44-x}\text{Au}_x\text{L}_{30}]^{4-}$ NCs produced by SME, we also compared their fragmentation habit with that of $[\text{Ag}_{44-x}\text{Au}_x\text{L}_{30}]^{4-}$ adopting conventional Au@Ag structure. The Au@Ag NCs were prepared by galvanic replacement reaction between $[\text{Ag}_{44}(\text{SR})_{30}]^{4-}$ NCs and Au(III) salt, yielding a mixture of $[\text{Ag}_{44-x}\text{Au}_x\text{L}_{30}]^{4-}$ ($x = 0-9$, see Supplementary Fig. 5). The most prominent $[\text{Ag}_{40}\text{Au}_4(\text{SR})_{30}]^{4-}$ ion was then subjected to MS/MS analysis (Fig. 4a), and its fragmentation pathways are summarized in Fig. 4b. Intriguingly, Au-core/Ag-shell $[\text{Ag}_{44-x}\text{Au}_x\text{L}_{30}]^{4-}$ (Fig. 4b) exhibits the same fragmentation behavior as $[\text{Ag}_{44}\text{L}_{30}]^{4-}$ (Supplementary Fig. 12), where the fragment cluster ions are successively developed by dissociation of L^- , $[\text{AgL}_2]^-$, and $[\text{Ag}_2\text{L}_3]^-$ from the parent or last-generation fragment cluster ions. This is in sharp contrast to the preferential dissociation of $[\text{AuL}_2]^-$ in the fragmentation process of $[\text{Ag}_{44-x}\text{Au}_x\text{L}_{30}]^{4-}$ NCs generated by SME, unambiguously manifesting the Ag@Au structure of the latter.”

Page 18, Figure 4:

Figure RL-2 is included in the main text as Figure 4.

2) Also, it will be great if some additional data such as XAS (EXAFS) can be included to directly unravel the Ag-Au structure.

Reply: Thanks for this constructive suggestion. We totally agree with the reviewer that the manuscript could be improved by additional evidences that could unambiguously support the Ag@Au structure of alloy NCs produced in SME reaction, among which X-ray absorption spectroscopy (XAS) analysis and resolving the cluster structure via single-crystal X-ray diffraction (XRD) are two very powerful techniques. It has been widely accepted that XAS analysis is useful for revealing the structural features of metal NCs/NPs, organometallic complexes and biomolecules (*Science* **2016**, 352, 797; *Nat. Commun.* **2015**, 6, 7664; *Nat. Commun.* **2016**, 7, 10414; *J. Am. Chem. Soc.* **2015**, 137, 7027), and especially effective in probing the local coordination environment of central atoms. However, due to the limited time and access to synchrotron radiation sources, a systematic investigation of bonding habit of Au heteroatoms in the Ag@Au NCs is out of the scope of the current study. This is definitely a good research topic of our follow-up study, which requires significant efforts on the systematic investigation. Alternatively, we believe the additional experiments we have conducted in this revision (as described in the coming paragraphs) could also address this issue.

In the present study, we communicated another powerful technology (tandem MS or MS/MS to reveal the location of heteroatoms in alloy NCs. The reviewers' insightful comments have prompted us to carry out more control experiments. In this revision, we are pleased to include another piece of direct

experimental evidence on the location of Au heteroatoms in alloy NCs. This evidence is from a structure comparison between present Ag@Au NCs and conventional Au@Ag NCs (Figures RL-1 and RL-2). The Au@Ag NCs were prepared by mixing $[\text{Ag}_{44}(\text{SR})_{30}]^{4-}$ NCs (SR denotes thiolate ligand) with Au(III) salt (i.e., HAuCl_4). The galvanic reduction of Au(III) salt by Ag(0) core of $[\text{Ag}_{44}(\text{SR})_{30}]^{4-}$ could extensively yield Au(0) heteroatoms, preferentially incorporating into the core of resultant alloy NCs. As can be seen in Figure RL-1, the as-produced alloy NCs by galvanic replacement reaction are a mixture of $[\text{Ag}_{44-x}\text{Au}_x(\text{SR})_{30}]^{4-}$ ($x = 0-9$) and other-sized NCs. Such Au@Ag $[\text{Ag}_{44-x}\text{Au}_x(\text{SR})_{30}]^{4-}$ NCs were then analyzed by tandem mass spectrometry (MS/MS).

The most prominent $[\text{Ag}_{40}\text{Au}_4(\text{SR})_{30}]^{4-}$ ion was then chosen as parent ions in the subsequent MS/MS analysis in a collision energy window of 5-30 eV. As shown in Figure RL-2, with elevating collision energy, 1st and 2nd generation of fragment cluster ions were generated by successive dissociation of single negatively charged SR⁻, $[\text{Ag}(\text{SR})_2]^-$ and $[\text{Ag}_2(\text{SR})_3]^-$ from the parent or last-generation fragment cluster ions. Such fragmentation pathways (Figure RL-2b) are identical to those of $[\text{Ag}_{44}(\text{SR})_{30}]^{4-}$ NCs (Supplementary Figure 12), but they are distinctly different from those of Ag@Au $[\text{Ag}_{44-x}\text{Au}_x(\text{SR})_{30}]^{4-}$ produced by the SME reaction. In the case of Ag@Au NCs, $[\text{Ag}_{44-x}\text{Au}_x(\text{SR})_{30}]^{4-}$, a preferential dissociation of $[\text{Au}(\text{SR})_2]^-$ rather than $[\text{Ag}(\text{SR})_2]^-$ or other Ag-containing modules are featured in the fragmentation pathways (Figure 3). Such deliberate comparison of MS/MS data unambiguously supports the Ag@Au structure of $[\text{Ag}_{44-x}\text{Au}_x(\text{SR})_{30}]^{4-}$ generated by the SME reaction developed in the present study.

We have included Figures RL-1 and RL-2 and necessary discussions in the revised manuscript.

Revisions:

SI, Page 6, Supplementary Figure 5:

Figure RL-1 is included in SI as Supplementary Figure 5.

Page 9, Lines 19-22:

“This set of UV-vis absorption and PAGE data is in good accordance to the ESI-MS analysis, which suggests AgAu NCs produced by galvanic replacement reaction are a mixture of $[\text{Au}_x\text{Ag}_{44-x}\text{L}_{30}]^{4-}$ ($x = 0-9$) and other-sized NCs (Supplementary Fig. 5).”

Page 17, Lines 7-19:

“To further confirm the Ag@Au structure of $[\text{Ag}_{44-x}\text{Au}_x\text{L}_{30}]^{4-}$ NCs produced by SME, we also compared their fragmentation habit with that of $[\text{Ag}_{44-x}\text{Au}_x\text{L}_{30}]^{4-}$ adopting conventional Au@Ag structure. The Au@Ag NCs were prepared by galvanic replacement reaction between $[\text{Ag}_{44}(\text{SR})_{30}]^{4-}$ NCs and Au(III) salt, yielding a mixture of $[\text{Ag}_{44-x}\text{Au}_x\text{L}_{30}]^{4-}$ ($x = 0-9$, see Supplementary Fig. 5). The most prominent $[\text{Ag}_{40}\text{Au}_4(\text{SR})_{30}]^{4-}$ ion was then subjected to MS/MS analysis (Fig. 4a), and its fragmentation pathways are summarized in Fig. 4b. Intriguingly, Au-core/Ag-shell $[\text{Ag}_{44-x}\text{Au}_x\text{L}_{30}]^{4-}$ (Fig. 4b) exhibits the same fragmentation behavior as $[\text{Ag}_{44}\text{L}_{30}]^{4-}$ (Supplementary Fig. 12), where the fragment cluster ions are successively developed by dissociation of L⁻, $[\text{AgL}_2]^-$, and $[\text{Ag}_2\text{L}_3]^-$ from the parent or last-generation fragment cluster ions. This is in sharp contrast to the preferential dissociation of $[\text{AuL}_2]^-$ in the fragmentation process of $[\text{Ag}_{44-x}\text{Au}_x\text{L}_{30}]^{4-}$ NCs generated by SME, unambiguously manifesting the Ag@Au structure of the latter.”

Page 18, Figure 4:

Figure RL-2 is included in the main text as Figure 4.

REVIEWERS' COMMENTS:

Reviewer #1 (Remarks to the Author):

I am satisfied with the changes made by the authors in the revised manuscript. My concerns raised for the original submission have been appropriately addressed. I can now recommend the publication of this work.

Reviewer #2 (Remarks to the Author):

The authors have revised the manuscript and responded adequately to all referee comments. I recommend the publication of this work in Nature Communications.

Reviewer #3 (Remarks to the Author):

The authors have addressed my questions satisfactorily, and the paper is now recommended for publication in the journal.